# A preliminary evaluation of FY-4A visible radiance data assimilation by the WRF (ARW v4.1.1)/DART (Manhattan release v9.8.0)-RTTOV (v12.3) system for a tropical storm case

Yongbo Zhou[1,2], Yubao Liu[1,2], Zhaoyang Huo[1,2], Yang Li[1,2]

[1]School of Atmospheric Physics, Nanjing University of Information Science and Technology, Nanjing, China

[2]Precision Regional Earth Modeling and Information Center (PREMIC), Nanjing University of Information Science and Technology, Nanjing, China

*Correspondence to*: Yongbo Zhou (yongbo.zhou@nuist.edu.cn), Yubao Liu (ybliu@nuist.edu.cn)

**Abstract**. Satellite visible radiance data that contain rich cloud and precipitation information are increasingly assimilated for improving forecasts of numerical weather prediction models. This study evaluates the Data Assimilation Research Testbed (DART, Manhattan release v9.8.0), coupled with the Weather Research and Forecasting (WRF) model (ARW v4.1.1) and the Radiative Transfer for TOVS (RTTOV, v12.3) package, for assimilating the simulated visible imagery of the FY-4A geostationary satellite located over Asia in an Observing System Simulation Experiment (OSSE) framework. The OSSE was performed for a Tropical Storm called Higos occurred in 2020 that contains multi-layer mixed-phase cloud and precipitation processes. The advantages and limitations of DART for assimilating FY-4A visible imagery were evaluated. Both single observation experiments and cycled DA experiments were performed to study the impact of different filter algorithms available in DART, variables being cycled, observation outlier thresholds, observation errors, and observation thinning.

The results show that assimilating visible radiance data significantly improves the analysis of cloud water path (CWP) and cloud coverage (CFC) from the first-guess forecasts. The Rank Histogram Filter (RHF) allows WRF to more accurately simulate CWP and CFC than the Ensemble Adjustment Kalman Filter (EAKF) although it took roughly twice time of the later. By cycling both cloud and non-cloud variables, specifying large outlier threshold values, or setting smaller observation errors without thinning of observations, WRF achieved better simulation of CWP and CFC. With model integration, DA of the visible radiance data also generated slightly positive impact on non-cloud variables as they were adjusted through the model dynamics and physics related to cloud processes. In addition, the DA improved the representation of precipitation. However, the impact on rain rate is limited by the inabilities of the DA to improve cloud vertical structures and cloud phases. Some negative impact of the DA on cloud variables was found due to the nature of the non-linear forward operator and the non-Gaussian distribution of the prior. Future works should explore faster and more accurate forward operators suitable for assimilating FY-4A visible imagery, techniques to reduce the non-linear and non-Gaussian errors, methods to correct the location errors which correspond to the underestimated clouds by the first guess, etc.

## 1. Introduction

All-sky satellite data assimilation (DA) has shown great potential to improve weather forecasts (Bauer et al., 2011). Many satellite DA-related studies were done for microwave (MW) and infrared (IR) radiance data. DA of MW radiance data adjusts the atmospheric state variables such as humidity and temperature (Geer et al., 2019; Migliorini and Candy, 2019) as well as cloud-related parameters such as cloud water/ice content and cloud coverage (Zhang et al., 2013; Yang et al., 2016), exhibiting positive effects on cloud and precipitation forecasting. All-sky MW data has been operationally assimilated at some Numerical Weather Prediction (NWP) centres (Bauer et al., 2010; Zhu et al., 2016). However, operational DA of all-sky MW data is limited to humidity- and temperature-sounding channels (Carminati and Migliorini, 2021) because MW radiance at these channels is insensitive to surface emissivity and skin temperature which are difficult to be estimated accurately under cloudy sky conditions (Hu et al., 2021a). DA of MW data is also challenging in terms of separating the radiance contribution of clouds from the non-cloud variables (especially temperature and humidity) (Geer et al., 2017). In addition, several existing studies showed positive effects on water vapour and temperature by assimilating the IR data in clear sky (McCarty et al., 2009; Ma et al., 2017). DA of IR radiance data in cloudy regions also improved the analysis of column integrated water and forecasting skills in the mid- and upper troposphere (Stengel et al., 2013; Geer et al., 2019). However, DA of IR radiance data in cloudy regions is still complicated by the non-linear relationship between the observation and state variables, the non-Gaussian problems (Li et al., 2022), the difficulty to separate cloud signals and non-cloud signals (Geer et al., 2017), and the inability to constrain the layered structures in the case of multi-layer clouds (Prates et al., 2014).

Several earlier studies suggested that there is great potential to assimilate the visible (VIS) and shortwave infrared (collectively referred to as shortwave, SW) data (Vukicevic et al., 2004; Polkinghorne and Vukicevic, 2011; Scheck et al., 2020; Schröttle et al., 2020) because these measurements contain some unique and supplementary cloud information to the IR and MW radiance data (Kostka et al., 2014; Schröttle et al., 2020). For example, SW radiation can penetrate a certain depth of cloud fields, and connotate cloud microphysical properties such as effective particle radius (Nakajima and King, 1990). In comparison, satellite IR data only provide information on the cloud top microphysics (Xue, 2009). SW data also complement precipitation radars that mainly measure large hydrometeors or precipitation particles (Keat et al., 2019) that usually do not occur during the initial growth stage of convective systems (Zhang and Fu, 2018), measuring mainly small cloud droplets. Furthermore, SW data usually have higher spatial resolution than MW data (Yang et al., 2017; Coste et al., 2017; Schimit et al., 2018). Therefore, high-resolution satellite SW radiance data provide cloud properties that are of great significance for cloud-resolving models. Unlike MW and IR data, the radiance contributed by clouds could be easily extracted from the VIS observations because the VIS radiance data is much more sensitive to cloud variables than non-cloud variables.

Several studies have attempted to assimilate the SW radiance data directly (i.e., direct DA). Unlike indirect DA which assimilates the retrieved cloud parameters from SW radiance data, Direct DA critically depends on observation operators.

Several observation operators and relevant algorithms have been developed for the satellite VIS radiance DA. For example, Vukicevic et al. (2004) mapped the model state variables to the equivalent radiance by an observation operator called the VIS and IR radiance measurements (VISIROO). Later on, Polkinghorne and Vukicevic (2011) developed the Spherical Harmonic Discrete Ordinate Method Plane Parallel for Data Assimilation (SHDOMPPDA), which solves radiative transfer processes by Discrete Ordinate Method (DOM) in Cartesian space while computing source functions using spherical harmonic series in spherical space. Compared with observation operators which solve source functions in Cartesian space, SHDOMPPDA has an advantage of high computation efficiency. To further speed up the computation, Scheck et al. (2016a) developed a Method for Fast Satellite Image Synthesis (MFASIS), which is 2 ~ 4 orders of magnitude faster than the other DOM-based observation operators (Scheck et al., 2016b). Scheck et al. (2018) further improved MFASIS and reduced its errors due to three-dimensional (3D) radiative effects. MFASIS is one of the observation operators in the Radiative Transfer for TOVS (RTTOV). RTTOV contains several observation operators for satellite radiance DA (Saunders et al., 2018), including DOM and the single-scattering method for SW radiative processes. These solvers could tackle cloud fraction, parallax correction, and many other critical aspects with respect to molecular absorption and scattering, underlying surface reflection, etc. (Saunders et al., 2018). Apart from these aforementioned observation operators, several machine learning-based observation operators and relevant methods (Scheck, 2021; Zhou et al., 2021) were developed and achieved high computation efficiency and accuracy for the VIS radiance simulations.

Another critical aspect of assimilating satellite VIS radiance data is in the selection of DA algorithms. There are two groups of mostly used DA approaches. The first is based on variational (VAR) methods. Vukicevic et al. (2004) assimilated GOES-9 VIS radiance data to the Regional Atmospheric Modeling System (RAMS) with a four-dimensional VAR (4DVAR) method and exhibited some positive effects on the short-term forecasting of a stratus cloud field. Similarly, Polkinghorne and Vukicevic (2011) assimilated the GOES-8 VIS and IR radiance data to RAMS with the 4DVAR system and also achieved some positive results. The second is based on ensemble-based methods. The ensemble-based methods are remarkably stable for nonlinear systems and were used for cloud and precipitation studies by several researchers (Lei et al., 2015; Kurzrock et al, 2019). Schröttle et al. (2020) assimilated VIS and IR radiance data in an idealized Observing System Simulation Experiment (OSSE) framework using a Local Ensemble Transform Kalman Filter (LETKF). Their results indicated that assimilating VIS radiance data alone could improve the forecasting skills of the regional model, COnsortium for Small-scale MOdeling (COSMO), and assimilating the VIS and IR radiance data collaboratively could further improve the forecasting skills. Their findings were further validated by Scheck et al. (2020) who concluded that assimilating the VIS radiance data of Spinning Enhanced Visible and Infrared Imager (SEVIRI) on METEOSAT could improve cloud and precipitation forecasts, and, meanwhile, reduce the temperature and relative humidity forecast errors in most conditions.

The VAR and ensemble-based approaches are complementary to each other. The ensemble-based approaches generate the flow-dependent background error covariance matrices which can be used to leverage the VAR approaches. Therefore, several hybrid approaches have been developed. Buehner et al. (2013) evaluated an ensemble-VAR DA approach by assimilating the observations that were operationally assimilated in Environment Canada, and found it more skilful than the VAR method for the short- and mid-range forecasts over tropical and extra-tropical regions. Gao et al. (2013) developed a

hybrid Ensemble Kalman Filter (EnKF)-3DVAR method to assimilate radar data and found it outperforms 3DVAR or EnKF
in shortening the spin-up time of a supercell storm. In addition, hybrid methods are increasingly applied in the DA of
satellite radiance data. Xu et al. (2016) assimilated the FY-3B satellite MW radiance data with the WRF hybrid
ensemble/3DVAR and improved the forecasts of typhoons' track, intensities, and precipitation from 3DVAR. Similar results
were also reported by Shen et al. (2020).

Nowadays, there are many community DA resources of the ensemble-based methods, such as the Data Assimilation
Research Testbed (DART, Anderson et al., 2009). DART supports several Numerical Weather Prediction (NWP) models
including the Weather Research and Forecasting (WRF) model (Skamarock et al., 2008). Recently, WRF/DART
incorporated the RTTOV radiative transfer package, facilitating the DA of satellite VIS to MW wavelengths radiance and
enabling the DA of all-sky satellite SW radiance data. The Advanced Geostationary Radiation Imager (AGRI) on the
geostationary FY-4A satellite launched in 2016, located over Asia, excels at high sampling frequency (5 min for intensive
observation and 15 min for operational observation) and high spatial resolution (0.5 ~ 2 km, depending on channels). Zhang
et al. (2019) pointed out that AGRI provides great radiance measurements describing rapidly evolving and small- to meso-
scale atmospheric systems. However, these data have not been assimilated in the operational NWP centres.

In this study, FY-4A/AGRI VIS radiance data were simulated in an OSSE framework and experiments of the VIS
radiance DA were conducted using the WRF/DART-RTTOV system. We intended to answer the following three questions.
1) What are the advantages and limitations of assimilating the FY-4A VIS radiance for forecasting tropical storms? 2) How
to specify the WRF/DART-RTTOV model settings and observations? 3) What needs to be considered for the real DA
applications of FY-4A VIS radiance data? The result of this study is preliminary toward evaluating the WRF/DART-RTTOV
system in assimilating all-sky FY-4A/AGRI VIS radiance data. It should be applicable to the DA of the upcoming FY-4B
VIS radiance data as well because the designs of the AGRI payload for the FY-4A and -4B are similar. The remaining
manuscript is organized as follows. The models and experiment designs are introduced in Section 2. The impact of
assimilating the FY-4A VIS radiance data on the analysis and first-guess forecasts of the tropical storm is discussed in
Section 3. Finally, the conclusion and an outlook of future works are summarized in Section 4.

## 2. Models and experiment designs

The OSSE framework in this study consists of a nature run, a control run, a cluster of single observation experiments, and a
group of cycled DA experiments. The nature run was performed to generate a proxy true atmosphere state. The DA
experiments that assimilated the simulated FY-4A VIS radiance data were carried out to explore the impact of the FY-4A
VIS radiance DA on the tropical storm forecast.  A control run that excluded DA was performed for comparison. The OSSE
was performed based on the tropical storm "Higos" in 2020.

On August 16, 2020, a tropical disturbance occurred over the north of Luzon, the Philippines. The system tracked
northwest toward the South China Sea, and intensified into a tropical storm on 19:00 August 17. The tropical storm was
further developed into a typhoon system named Higos on 12:00 UTC, August 18. Higos landed on Zhuhai, Guangdong

province on 22:00 UTC, August 18, and weakened into a tropical depression on 12:00 UTC, August 19. This study focuses on the pre-landfall stage of the tropical storm (00:00 UTC ~ 12:00 UTC, August 18) in consideration that FY-4A VIS imagery is only available at daytime. During this period, the tropical storm had multi-layer and mixed-phase cloud structures, which facilitates the evaluation of the abilities of assimilating VIS radiance data for these cloud structures.

## 2.1 Configurations of the WRF Mode

The WRF model domain settings were kept same for the nature run, control run, and DA experiments to avoid errors due to displacement of grids between the observation and simulations. The WRF model domain covers parts of the East Asia and Western Pacific (Figure 1). The domain contains $151 \times 177$ horizontal grid boxes with a grid spacing of 15 km in the horizontal directions and 40 vertical levels, and the model top was set to 50 hPa. To avoid the disturbances over the regions close to the model domain boundaries, simulations within the inner rectangle of $131 \times 157$ horizontal grids are analysed.

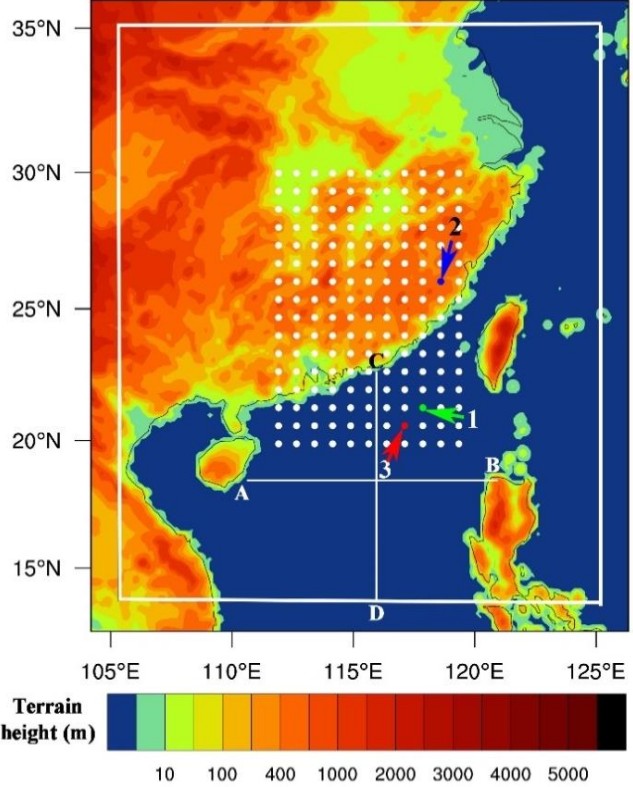

**Figure 1. The WRF model domain with 15-km horizontal grid spacing. Only the observations within the inner white rectangle were assimilated and those close to the model domain boundaries were discarded. The white dots denote the locations that were used for single observation experiments (discussed in Section 2.3.2). Detailed discussions are given for points 1 (green), 2 (blue), and 3 (red) in the text. The two white lines, AB and CD, are for cross-section analyses.**

For the nature run, the initial conditions (ICs) and lateral boundary conditions (LBCs) were extracted from the National Centers for Environmental Prediction (NCEP) Final (FNL) Operational Global Analysis data (with $1° \times 1°$ resolution,

available at https://rda.ucar.edu/datasets/ds083.2/). The WRF model configurations include the Thompson microphysical
scheme (Thompson et al., 2008), the Tiedtke Cumulus Parameterization option (Tiedtke, 1989; Zhang et al., 2011), and the
University of Washington (UW) planetary boundary layer scheme (Bretherton and Park, 2009). These are the optimal
schemes for typhoon simulations over the Northwest Pacific Ocean as suggested by Di et al (2019). The other model
configurations include the revised MM5 Monin-Obukhov surface layer scheme (Jiménez et al., 2012), the five layer thermal
diffusion land surface scheme (Dudhia, 1996), and the Rapid Radiative Transfer Model for Global Climate Models (RRTMG)
longwave and shortwave radiation schemes (Iacono et al., 2008). The Thompson microphysical scheme provides prognostic
variables of liquid water particles including cloud water droplets and rain drops, and ice particles including ice crystals, snow,
and graupel. The nature run was initialized with a cold start at 12:00 UTC 17 August 2020. To exclude a spin-up time of 14
hr, the WRF model simulations between 02:00 and 12:00 UTC, 18 August 2020 were used as a proxy true atmosphere of the
tropical storm. The nature run captured the track and general properties of Higos. Synthetic observations of FY-4A VIS
radiance were simulated with RTTOV which will be described in details in Section 2.2. The simulated VIS imagery (15 km
$\times$ 15 km resolution) was approximately equivalent to the superobbed 2km-resolution imagery (as those provided by FY-4A
real observations) by averaging the 2 km $\times$ 2 km imagery for every block of about $7 \times 7$ pixels. Because the observation
locations and model grid points are overlapped, the locations of the synthetic observations are directly assigned to the model
grid points without interpolation during the DA processes.

For the cycled DA experiment, the ensemble size is set to 40 and the ICs and LBCs were extracted from the ERA5
hourly data at $0.25°\times0.25°$ resolution (available at https://cds.climate.copernicus.eu/api/v2). Perturbations, which were
extracted based on the WRF 3DVAR system using a generic background error option with proper scaling, were added to the
ICs. The scaling factors for the variance, horizontal length scale, and vertical length scale are set to 0.25, 1.0, and 1.5,
respectively. To avoid discontinuities and poor results at the boundary, LBCs at each analysis time were updated based on
the analysis and WRF lateral boundary conditions using an approach built in the DART *pert_wrf_bc* module. This is the
reason why we choose the higher resolution LBCs, as will be done in real DA applications, for the DA experiments than for
the nature run. The WRF model microphysics configurations are the same as the nature run. The ensemble members were
initialized by a cold start at 00:00 UTC 18 August 2020. After a spin up of 2 hr, synthetic visible radiance observations were
assimilated to the ensemble members from 02:00 to 09:00 UTC 18 August 2020. The time span corresponds to the daytime
when VIS imagery is available. The ensemble members were advanced to 12:00 UTC. The updating frequency of the first-
guess state variables was set according to different experiment designs summarized in Table 1. With these set ups, the effects
of assimilating VIS imagery on the spin up of the WRF model and on the analysis and first-guess forecasts of the state
variables including cloud and precipitation were explored. The model settings for the control run are the same as the cycled
DA experiments, except that no observations were assimilated.

## 2.2 Configurations of the RTTOV radiative transfer package

Synthetic AGRI channel 2 radiance was simulated based on WRF outputs using the RTTOV radiative transfer package. The input parameters of RTTOV include cloud-related parameters (the vertical structures of liquid water mixing ratio, ice water mixing ratio, cloud water effective radius, cloud fraction, etc.), atmosphere profiles (the water vapour mixing ratio profile, temperature profile, etc.), surface properties (elevation, surface type, etc.), sun-satellite viewing geometries, etc. In the partly cloudy regions, the top-of-atmosphere (TOA) radiance is a weighted average of the radiance for a clear sky and a cloudy sky. The weight attached to the cloudy sky (i.e., the effective cloud fraction) was calculated as the vertically averaged cloud fraction weighted by the mixing ratio of hydrometeors (Geer et al., 2009). It is noted that cloud fraction (CFC) parameterization in WRF model depends on relative humidity (RH), the saturation water vapour mixing ratio ($q^*$), and cloud water + ice mixing ratios ($q_{l+i}$) (Xu and Randall, 1996),

$$\text{CFC} = \begin{cases} RH^p \left[ 1 - \exp\left( \frac{-\alpha q_{l+i}}{[(1-RH)q^*]^\gamma} \right) \right], & if\ RH < 1 \\ 1, & if\ RH \geq 1 \end{cases} \tag{1}$$

where $p$, $\alpha$, and $\gamma$ are suggested to be 0.25, 100, and 0.49 separately.

The solar zenith angle, solar azimuth angle, satellite viewing zenith angle, and satellite azimuth angle were calculated using the Python *astropy* library according to the UTC time and the FY-4A satellite position (104.7 °E). In addition, errors ranging from 1 ~ 4 mW m$^{-2}$ sr$^{-1}$ were assigned to the observations (Table 1). RTTOV includes the schemes considering different pre-defined cloud optical properties. For liquid water clouds, the "Deff" scheme where cloud optical properties are parameterized in terms of Re (Mayer and Kylling, 2005) was used. The cirrus scheme developed by Baren et al. (2014) was used to calculate ice cloud optical properties, which has no explicit dependence on ice particle size. Therefore, analyses of the results were simplified given that cloud variables were adjusted collectively but we do not have to analyse the effective radius of ice particles.

The radiative transfer processes were simulated by the DOM solver in RTTOV. The surface was treated as a specular reflector for downwelling emitted radiance. For land surface, the surface Bidirectional Reflectance Distribution Function (BRDF) was drawn from the land surface atlases (Vidot and Borbás, 2014; Vidot et al., 2018). For sea surface, BRDF was calculated with the JONSWAP (Hasselmann et al., 1973) solar sea BRDF model. The lay-to-space transmittance was computed by the v9 predictor on 54 levels (Matricardi, 2008). The downwelling atmospheric emission was computed using the linear-in-tau approximation for the Planck source term. Water vapour profiles were drawn from the WRF outputs. Other parameters not explicitly mentioned are set to the default values in DART/RTTOV.

Based on the above model configurations, the dependence of AGRI channel 2 radiance on the Cloud Water Path (CWP) and effective radius of cloud water droplet (Re) is presented in Figure 2. CWP denotes the vertically integrated cloud liquid and ice water mixing ratio in an atmospheric column, which is calculated by,

$$\text{CWP} = \int_{P_s}^{P_t} \frac{1}{g} (Q_c + Q_i) dP \tag{2}$$

where $P_s$ and $P_t$ denote the surface and model top pressures. Qc and Qi are the liquid water mixing ratio (the sum of the mixing ratio of cloud droplet and rain) and ice water mixing ratio (the sum of the mixing ratio of ice, snow, and graupel), and $g$ the gravitational acceleration (9.8 ms$^{-2}$).

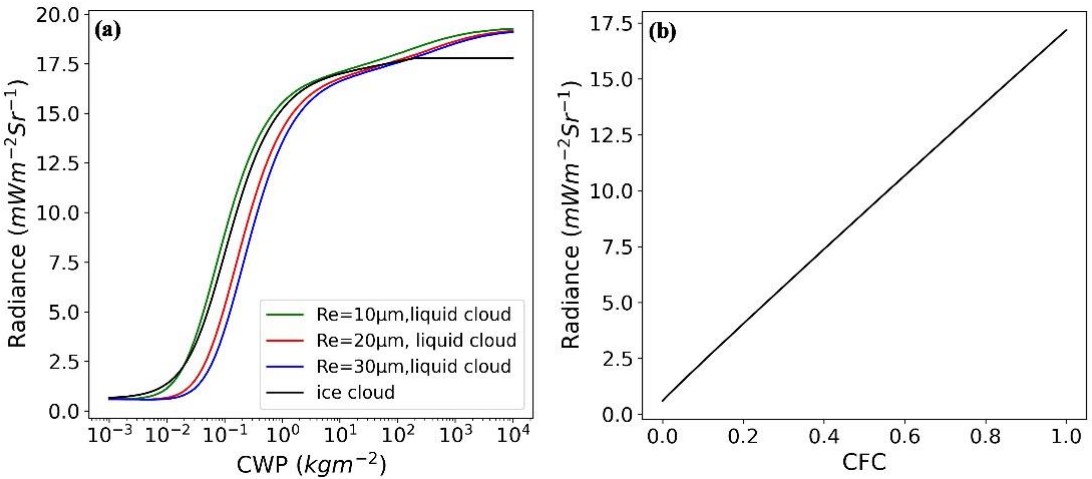

Figure 2. Dependence of AGRI channel 2 radiance on (a) cloud water path (CWP) and effective radius (Re), and (b) cloud fraction (CFC) for CWP of 10 kg m$^{-2}$ and Re of 15 μm. The simulation was performed with the "Deff" scheme for liquid water cloud optical properties and the Baren et al. (2014) scheme for cirrus optical properties. The solar zenith angle, viewing zenith angle, and relative azimuth angle were set to 25 °, 40 °, and 135 °, respectively.

The curvature properties in Figure 2 indicate a non-linear relationship between the observation (radiance) and cloud parameters (CWP and Re). The variation of the radiance-CWP functions with different effective radii become smaller as Re increases. For Re larger than 30 μm, the radiance-CWP functions almost do not change with effective radii. Because raindrops are several orders larger than cloud droplets, the effective radius of cloud droplets is sufficient to describe the radiative transfer processes for the clouds where cloud droplets and raindrops coexist. As a result, Re in the following discussion explicitly denotes the effective radius of cloud droplets, which corresponds to the WRF state variable "RE_CLOUD".

## 2.3 DA experiment design and DART configurations

### 2.3.1 DART filters

DART was configured to employ the Ensemble Adjustment Kalman Filter (EAKF, Anderson, 2001) and the Rank Histogram Filter (RHF, Anderson, 2010) for this study. EAKF and RHF are two variants of the deterministic filters. Therefore, no perturbations were added to the observations. EAKF is a serial ensemble DA algorithm and the observations are assimilated as scalars. The model state variable $x_m$ is updated by Equation (3) (Anderson, 2001),

$$x'_m = x_m + \Delta x_{m,n}, \qquad m = 1, \dots, M, n = 1, \dots, N \tag{3}$$

where $x_m$ denotes the m[th] state variable, $x'_m$ the updated value of $x_m$, and $\Delta\mathbf{x}_{m,n}$ the state variable increment for the m[th] state variable due to the n[th] observation. $\Delta\mathbf{x}_{m,n}$ is calculated by Equation (4),

$$\Delta\mathbf{x}_{m,n} = \left(\sigma_{p,m}/\sigma_p^2\right)\Delta y_n, \qquad m = 1, \dots, M, n = 1, \dots, N \tag{4}$$

where the subscript "$p$" stands for the prior estimate (i.e., the first guess), $\sigma_{p,m}$ is the first-guess sample error covariance between the observation and the m[th] state variable $x_m$, and $\sigma_p^2$ the first-guess sample error variance of the observed variable. $\Delta y_n$ is the observation increment for the n[th] ensemble, which is calculated by the following equation,

$$\Delta y_n = \left(y_n^p - \bar{y}_p\right)\left(\sigma_u/\sigma_p\right) + \bar{y}_u - y_n^p, \qquad n = 1, \dots, N \tag{5}$$

where $y_n^p$ denotes the first guess of the observed variable for the n[th] ensemble, $\bar{y}_p$ the ensemble mean of the first guess of the observed variable, $\bar{y}_u$ the ensemble mean of the posterior estimate (i.e., the analysis) of the observed variable, $\sigma_u$ the updated standard deviation of $\sigma_p$. $\bar{y}_u$ and $\sigma_u$ are calculated by Equations (6) ~ (7).

$$\overline{y_u} = \frac{\sigma_o^2}{\sigma_o^2 + \sigma_p^2}\bar{y}_p + \frac{\sigma_p^2}{\sigma_o^2 + \sigma_p^2}y_o \tag{6}$$

$$\sigma_u = \frac{\sigma_o \sigma_p}{\sqrt{\sigma_o^2 + \sigma_p^2}} \tag{7}$$

where $y_o$ and $\sigma_o$ denote the observation and its corresponding observational error standard deviation.

Anderson (2007; 2009) promoted a spatially varying state-space adaptive covariance inflation to the first-guess state to increase the prior ensemble spread. The same option was adopted in this study and also several other papers (Lei et al., 2015; Kurzrock et al., 2019). The adaptive inflation uses 1.0, 0.6, and 0.9 as the initial value, fixed standard deviation, and damping settings, respectively. The sampling error due to the use of the limited ensemble size was corrected by the method developed by Anderson (2012). Since observations like satellite VIS radiance data do not have a specific single vertical location, no vertical localization was used in this study.

The RHF produces a posterior ensemble based on a continuous approximation of the prior Probability Density Function (PDF) and a piecewise linear representation of the likelihood. The prior PDF is approximated by a rank histogram which has a piecewise constant between two ensemble members and follows Gaussian distributions beyond the lower and upper bounds of the ensemble members. The posterior distribution is calculated by the Bayes Theorem, and the state variable is updated by searching the appropriate position in the state variable space which partitions the posterior distribution to unity probability for each ensemble member. The prior PDF does not have to respect the Gaussian form for RHF. Therefore, the method is declared to be more suitable for non-Gaussian problems. Details on this algorithm can be found in Anderson (2010).

### 2.3.2 Single observation experiments

With the OSSE setups, a set of single observation experiments were performed by employing EAKF to assimilate the VIS radiance data. The single observation experiments assimilate an observation at a given pixel, and the adjustment of state variables in the column at a targeting pixel is only caused by assimilating the one observation. This is convenient to evaluate the basic functionality of assimilating VIS radiance data. The potentials, inabilities and ambiguities of assimilating VIS radiance data were discussed herein. The single observation experiments were performed at 02:00 UTC 18 August 2020, and 265 no forecast was carried out. The affected cloud variables include Qc, Qi, Re, CFC, and the non-cloud variables including water vapour mixing ratio (Q), perturbation potential temperature (T), and the x- and y-wind components (U and V).

The single observation experiments were performed for the most inner parts of the satellite imagery to avoid disturbances near boundaries. The observations at 02:00 UTC were thinned by selecting every six pixels to make sure that the selected observations are far from each other. This resulted in 176 points shown in Figure 1. By setting a localization 270 distance of 15 km, assimilating the VIS radiance at a pixel would not influence the state variables at the surrounding pixels. Therefore, we performed a cluster of 176 single observation experiments in one DA cycle to save computational cost, similar to Scheck et al. (2020). Amongst the 176 selected pixels, special focuses were given to the three coloured points, which were designed to illustrate the ambiguities related to cloud layered structures and cloud phases and to illustrate the limitations due to the non-Gaussian and non-linear problems.

### 275 2.3.3 Cycled DA experiments

Fourteen cycled DA experiments were performed to evaluate the influences of different model settings and observation pre-processing on the analysis and forecast with VIS radiance DA. The purpose of the cycled DA experiments is to reveal the forecast quality and growth of the forecasting errors during assimilating satellite VIS radiance data, and to provide some guidance on the use of WRF/DART-RTTOV and an outlook to the future applications. The experiment setups cover the tests 280 of different filter algorithms, cycling intervals, cycling variables, outlier threshold values, observation errors, and observations with or without data-thinning. The outlier threshold value for the observation is a pre-defined threshold for rejecting an observation depending on its distance from the ensemble mean of the first guess. If the distance is more than N (the predefined outlier threshold value) standard deviations from the square root of the sum of the first-guess ensemble and observation error variance, the observation is rejected. A description of the experiment designs are summarized in Table 1.

Comparison between Exp-01 ~ Exp-03 and Exp-4~Exp-06 groups was designed to reveal the pros and cons of EAKF and RHF on the WRF analysis and forecast. Comparison between Exp-01 ~ Exp-02 and Exp-07 ~ Exp-08 groups was designed to reveal the influences of updating the thermal and dynamic variables. Comparison between Exp-03, Exp-09, and Exp-10 was designed to reveal the influences of the observation errors. Comparison between Exp-01 ~ Exp-03 and Exp-11~ Exp-13 groups was designed to reveal the influences of the outlier threshold values. Finally, comparison between Exp-10 290 and Exp-14 was designed to reveal the influences of observation thinning.

**Table 1.** Parameter settings for the cycled data assimilation experiments. $x_{cloud}$ denotes the WRF cloud variables including cloud fraction (CLDFRA), the mixing ratio of cloud droplet (QCLOUD), rain (QRAIN), ice (QICE), snow (QSNOW), graupel (QGRAUP), the effective radius of cloud water droplet (RE_CLOUD), and the effective radius of cloud ice droplet (RE_ICE). $x_{atmos}$ denotes the WRF non-cloud variables including water vapour mixing ratio (QVAPOR), water vapour mixing ratio at 2 m height (Q2), x-, y-, and z-wind components (U, V, W), x- and y-wind components at 10 m height (U10 and V10), temperature at 2 m height (T2), perturbation geopotential (PH), perturbation potential temperature (T), perturbation dry air mass in column (MU), and surface pressure (PSFC).

| DA experiments | Thinning length | Localization distance | Filter algorithm | Cycling interval | Cycling variables | Outlier threshold | Observation error |
|---|---|---|---|---|---|---|---|
| Exp-01 | —— | 15 km | EAKF | 10 min | $x_{cloud}+x_{atmos}$ | 3 | 1 mW m$^{-2}$ Sr$^{-1}$ |
| Exp-02 | —— | 15 km | EAKF | 1 hr | $x_{cloud}+x_{atmos}$ | 3 | 1 mW m$^{-2}$ Sr$^{-1}$ |
| Exp-03 | —— | 15 km | EAKF | 3 hr | $x_{cloud}+x_{atmos}$ | 3 | 1 mW m$^{-2}$ Sr$^{-1}$ |
| Exp-04 | —— | 15 km | RHF | 10 min | $x_{cloud}+x_{atmos}$ | 3 | 1 mW m$^{-2}$ Sr$^{-1}$ |
| Exp-05 | —— | 15 km | RHF | 1 hr | $x_{cloud}+x_{atmos}$ | 3 | 1 mW m$^{-2}$ Sr$^{-1}$ |
| Exp-06 | —— | 15 km | RHF | 3 hr | $x_{cloud}+x_{atmos}$ | 3 | 1 mW m$^{-2}$ Sr$^{-1}$ |
| Exp-07 | —— | 15 km | EAKF | 10 min | $x_{cloud}$ | 3 | 1 mW m$^{-2}$ Sr$^{-1}$ |
| Exp-08 | —— | 15 km | EAKF | 1 hr | $x_{cloud}$ | 3 | 1 mW m$^{-2}$ Sr$^{-1}$ |
| Exp-09 | —— | 15 km | EAKF | 3 hr | $x_{cloud}+x_{atmos}$ | 3 | 2 mW m$^{-2}$ Sr$^{-1}$ |
| Exp-10 | —— | 15 km | EAKF | 3 hr | $x_{cloud}+x_{atmos}$ | 3 | 4 mW m$^{-2}$ Sr$^{-1}$ |
| Exp-11 | —— | 15 km | EAKF | 10 min | $x_{cloud}+x_{atmos}$ | 6 | 1 mW m$^{-2}$ Sr$^{-1}$ |
| Exp-12 | —— | 15 km | EAKF | 1 hr | $x_{cloud}+x_{atmos}$ | 6 | 1 mW m$^{-2}$ Sr$^{-1}$ |
| Exp-13 | —— | 15 km | EAKF | 3 hr | $x_{cloud}+x_{atmos}$ | 6 | 1 mW m$^{-2}$ Sr$^{-1}$ |
| Exp-14 | 60 km | 60 km | EAKF | 3 hr | $x_{cloud}+x_{atmos}$ | 3 | 1 mW m$^{-2}$ Sr$^{-1}$ |

## 2.4 Metrics of simulation errors

Root Mean Square Error (RMSE) and Mean Absolute Error (MAE) are two of the most used metrics to assess the weather simulation errors (Kurzrock et al., 2019). RMSE is much more sensitive to extremely large errors than MAE. For satellite VIS radiance DA, some extremely large analysis increments of CWP were rarely expected (details provided in Section 3), implying that the difference of RMSE between the first guess and the analysis was not as distinct as MAE. Thus, MAE is used to measure the difference between the simulated CWP and the theoretical true CWP (derived from the nature run). MAE is calculated by the following formula,

$$\text{MAE} = \frac{1}{n_x n_y} \sum_{i,j} abs(x_{i,j}^{\text{sim}} - x_{i,j}^{obs}) \tag{8}$$

where $x_{i,j}^{\text{sim}}$ ($x_{i,j}^{obs}$) denotes the simulated (true) CWP at the $i^{\text{th}}$ (in the zonal direction) and $j^{\text{th}}$ (in the meridional direction) model grid. $n_x$ and $n_y$ denote the number of pixels in zonal and meridional directions of the relevant model domains.

Fraction skill score (FSS) was developed to measure the accuracy of a spatially inhomogeneous variables on a certain spatial scale. Therefore, it can mitigate the "double-penalty" problem (Mittermaier et al., 2013) for small spatial shifts of interested features. FSS is calculated by the following formula,

$$FSS = 1 - \frac{\frac{1}{m_x m_y}\Sigma_{i,j}\left(p_{i,j}^{obs}-p_{i,j}^{sim}\right)^2}{\frac{1}{m_x m_y}(\Sigma_{i,j}\, p_{i,j}^{obs}+\Sigma_{i,j}\, p_{i,j}^{sim})} \tag{9}$$

where $p_{i,j}^{obs}$ denotes the cloud fraction within a subdomain covering $3\times3$ model grids. $m_x$ and $m_y$ denote the dimensions of subdomains in the zonal and meridional directions.

For precipitation simulation evaluation, Threat Score (TS) is used. TS is computed as,

$$TS = \frac{H}{F+M+H} \tag{10}$$

where $H$ denotes the number of pixels with the correct representation of precipitation (hits), $F$ denotes the number of pixels where simulation indicates precipitation while the true state indicates non-precipitation (false alarms), and $M$ denotes the number of pixels where simulation indicates non-precipitation while the true state indicates precipitation (under predictions).

Following Scheck et al. (2020), we also use the Mean Profile Error (MPE, $\varepsilon$) to assess the error of model state with respect to the nature run. If the difference between the MAE of the analysis ($\varepsilon_{pos}$) and that of the first guess ($\varepsilon_{pri}$), calculated as $\delta\varepsilon = \varepsilon_{pos} - \varepsilon_{pri}$, is negative, positive impact is generated by the DA procedure, and vice versa.

## 3. Results

### 3.1 Single observation experiments

The results discussed herein correspond to the OSSE setups described in Section 2.3.2. Only the ensemble mean of the first guess and analysis of state variables were discussed. We focused on three cases. 1) In both state variable (or a diagnosed parameter such as CWP) and observation spaces, the analysis is within the range bounded by the first guess and the truth; 2) The analysis is within the first guess and the truth in the observation space but not in the state variable space. This case is associated with the spurious covariance and non-linear properties of the forward operator; 3) The analysis is beyond the first guess and the truth in both observation space and state variable space, which is closely related to the non-Gaussian properties of the prior PDF. The results including the three cases are shown in Figure 3.

Assimilating VIS radiance data generated trivial impact on the non-cloud variables including x- and y-wind components (U and V in Figure 3(a)), temperature and water vapour mixing ratio (T and Q in Figure 3(b)). From the perspective of radiative transfer, the VIS radiance is insensitive to U and V at the analysis time. Therefore, the adjustment in observation space should not influence U and V. In addition, the VIS radiance is closely related to CFC (Figure 2(b)), an implicit relationship between the VIS radiance and RH could be expected because the parameterization of CFC involves water vapor (Equation (1)). The VIS radiance is positively related to CFC. However, given that RH depends not only on Q, but also on T

and pressure, spurious covariance between the VIS radiance and Q/T may be generated due to the ensemble spread of Q/T. The ensemble spread of Q/T for the ensemble members would blur the relationship between VIS radiance and Q/T.

Therefore, only neutral impact on Q and T was revealed for the single observation experiments.

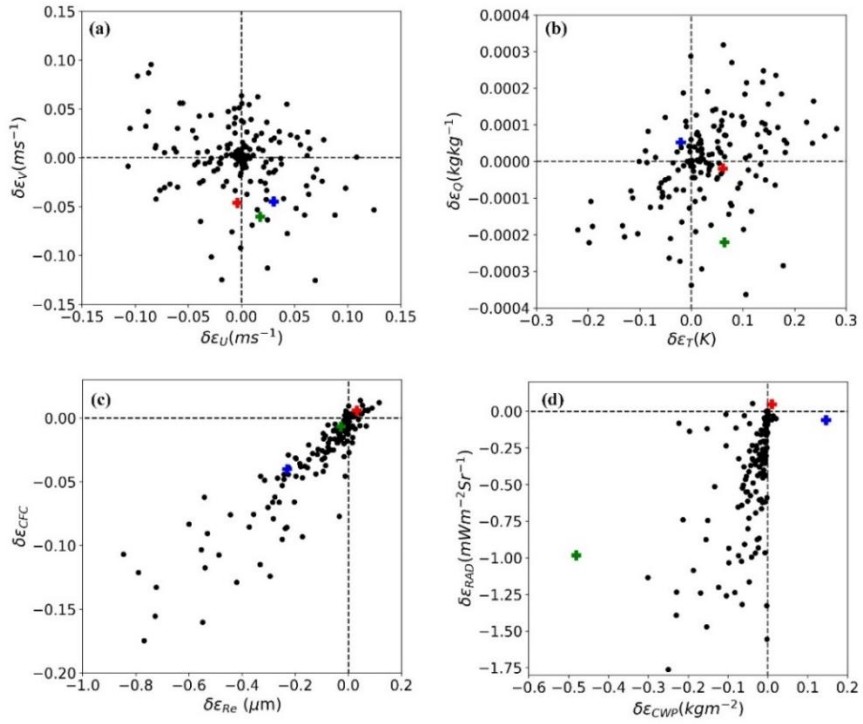

**Figure 3. Differences of the Mean Profile Error (MPE) between the ensemble mean of the first guess and analysis, denoted as** $\delta\varepsilon_X = X_{MPE,pos} - X_{MPE,pri}$**, where X denotes a state variable or a diagnosed parameter, and** *pos* **and** *pri* **the analysis and first guess, respectively. The variables include (a) the x- and y-wind components (U and V), (b) the perturbation potential temperature (T)**
**and water vapour mixing ratio (Q), (c) the cloud fraction (CFC) and effective radius (Re), and (d) cloud water path (CWP) and radiance (RAD). The plus signs in green, blue, and red colours correspond to points 1, 2, and 3 in Figure 1.**

Assimilating VIS radiance data improved the ensemble mean of Re, CFC, CWP, and VIS radiance for most points in Figure 3(c) and 3(d). Take Point 1 marked in Figure 1 as an example for case 1. The profiles of the ensemble mean of cloud and non-cloud variables for the first guess and analysis are shown in Figure 4. Point 1 corresponds to a single ice cloud layer

between 400 ~ 200 hPa with a CWP of 0.01 kg m$^{-2}$ and a TOA radiance of 3.63 mW m$^{-2}$ Sr$^{-1}$ for the nature run. The first guess features a two-layer mixed-phase cloud with a false alarm liquid water cloud layer below 500 hPa. The ensemble mean first-guess CWP and the equivalent VIS radiance are 1.33 kg m$^{-2}$ and 7.29 mW m$^{-2}$ Sr$^{-1}$, respectively. After assimilating the satellite VIS radiance data, the first guess was drawn toward the nature run both in the observation space, with a decreased ensemble mean radiance of 6.40 mW m$^{-2}$ Sr$^{-1}$, and in CWP, with a decreased ensemble mean CWP of 0.85 kg m$^{-2}$. As a

result, Qc, Qi, CFC, and Re were adjusted collaboratively toward the nature run.

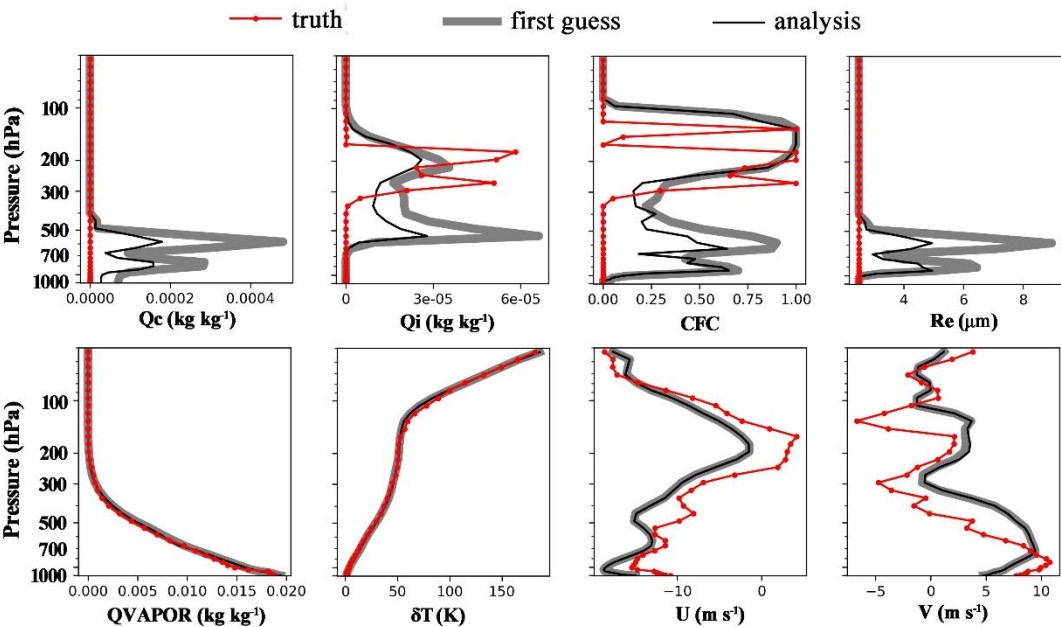

**Figure 4. The vertical profiles of state variables for the nature run (theoretical truth), the first guess and analysis (ensemble mean). Qc denotes the liquid water mixing ratio, Qi the ice water mixing ratio, CFC the cloud fraction, Re the effective radius of liquid water droplets, QVAPOR the water vapour mixing ratio, δT the perturbation potential temperature, U and V the x- and y-wind components.**

Since the simulated VIS radiance observation is not sensitive to cloud vertical structures but to the accumulated cloud water/ice mass, assimilating the VIS radiance observation could not reduce cloud vertical location errors. In addition, it could not remove the false alarm liquid clouds produced by the spurious covariance between the VIS radiances and liquid water clouds in the background. According to Equation (4), the analysis increment of each state variable is linearly related to its covariance with observations. Therefore, the vertical structures and hydrometeor phases of the analysis are mainly determined by those of the first guess. A larger first guess of the state variable would generate larger covariance, and a larger adjustment to the first guess would be expected. Because the ensemble mean of the first-guess Qc and Qi were larger in the lower layer ($\geq$ 400 hPa), the adjustment of Qc/Qi were much more distinct for the lower layer than the upper layer ($\leq$ 300 hPa). Similar results were also found for Re except that larger liquid water particles occurred in the middle layer (~ 600 hPa) and smaller liquid water particles occurred in the lower layer (~ 800 hPa). The covariance between CFC and the synthetic observation is zero in the upper layer ($\leq$ 200 hPa) because CFC is almost a constant of 1 for all ensemble members (the ensemble spread of CFC is zero). Compared with the cloud variables, the non-cloud variables remain almost unchanged after the DA.

Case 2 is used to illustrate that positive impact on VIS radiance does not ensure positive impact on CWP. Take Point 2 in Figure 1 as an example, the theoretical truth, the ensemble mean of the first guess and the analysis of radiance/CWP are 8.59 mW m$^{-2}$ Sr$^{-1}$/0.17 kg m$^{-2}$, 7.94 mW m$^{-2}$ Sr$^{-1}$/2.50 kg m$^{-2}$, and 8.00 mW m$^{-2}$ Sr$^{-1}$/2.65 kg m$^{-2}$, respectively. That is to say,

the analysis of CWP is beyond the range bounded by the first guess and the truth. This is partly due to the non-linear relationship between CWP and VIS radiance. To elaborate this problem, we calculated the ensemble mean with Formula (4) and substituting $\overline{y_u}$ in Formula (6) would get the following formula,

$$380 \quad \overline{\Delta x_m} = \frac{\sigma_{p,m}}{\sigma_o^2 + \sigma_p^2} R_{inc} \tag{11}$$

where $\overline{\Delta x_m}$ denotes the ensemble mean of the $m^{th}$ state variable increment, $R_{inc}$ the ensemble mean radiance increment, which is calculated by the following formula,

$$R_{inc} = y_o - \bar{y}_p \tag{12}$$

Considering a simplified case with just 2 ensemble members, the ensemble mean observation increment is calculated by
385 the following formula,

$$R_{inc} = F(W_o) - \frac{F(W_1) + F(W_2)}{2} \tag{13}$$

where $W_o$ denotes the observed CWP and $F$ denotes the forward operator. $W_1$ and $W_2$ represent CWP of the two ensemble members. However, considering the relationship between CWP and the VIS radiance, the theoretical true observation increment should be,

$$390 \quad R_{inc}^t = F(W_o) - F(\overline{W}), \overline{W} = \frac{W_1 + W_2}{2} \tag{14}$$

As indicated by Figure 4, $R_{inc}$ is larger than $R_{inc}^t$. That is, the ensemble mean observation increment was overestimated by Equation (14), leading to an over-estimated ensemble mean of the analysis of CWP.

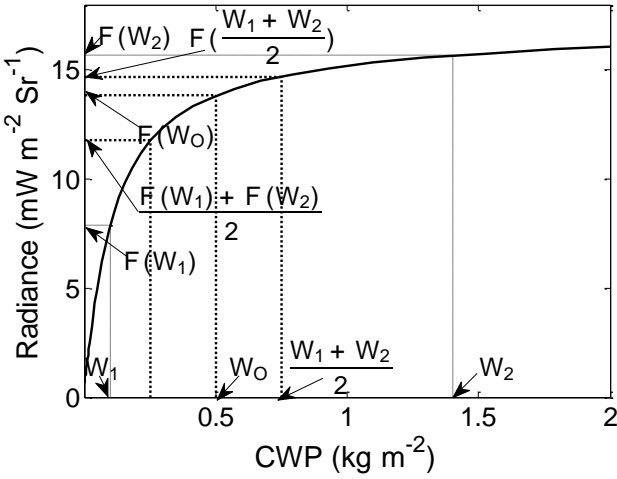

Figure 5. Illustration of the nonlinear effects of the observation operator on the calculation of radiance increments with 2 ensemble
members. F denotes the observation operator, $W_1$ and $W_2$ denote cloud water path (CWP) for the 1st and 2nd ensemble member,
$W_o$ denotes the observed CWP.

Case 3 is to show that some negative impact could be generated in the observation space in some conditions. Take Point 3 in Figure 1 as an example, the ensemble mean radiance (2.51 mW m$^{-2}$ Sr$^{-1}$) of the analysis is beyond the range bounded by the ensemble mean of the first guess (2.56 mW m$^{-2}$ Sr$^{-1}$) and the true radiance (3.41 mW m$^{-2}$ Sr$^{-1}$). The EAKF algorithm assumes that the prior PDF, $p(x)$, of the equivalent observations and model state variables (or the diagnosed variable such as CWP in this study) confirms to Gaussian functions. To see how well the assumption was respected, $p(x)$ of radiance and CWP is presented in Figure 6. It shows non-Gaussian prior PDFs. Several studies concluded that the non-Gaussian properties negatively affect the performance of ensemble methods (Lawson and Hansen, 2004; Lei et al., 2010). Therefore, we tentatively ascribe the negative impact in the observation space partly to the non-Gaussian properties of $p(x)$. Accordingly, DA experiments using RHF, which is not bound by the assumption of Gaussian, were added to the cycled DA experiments in comparison with EAKF.

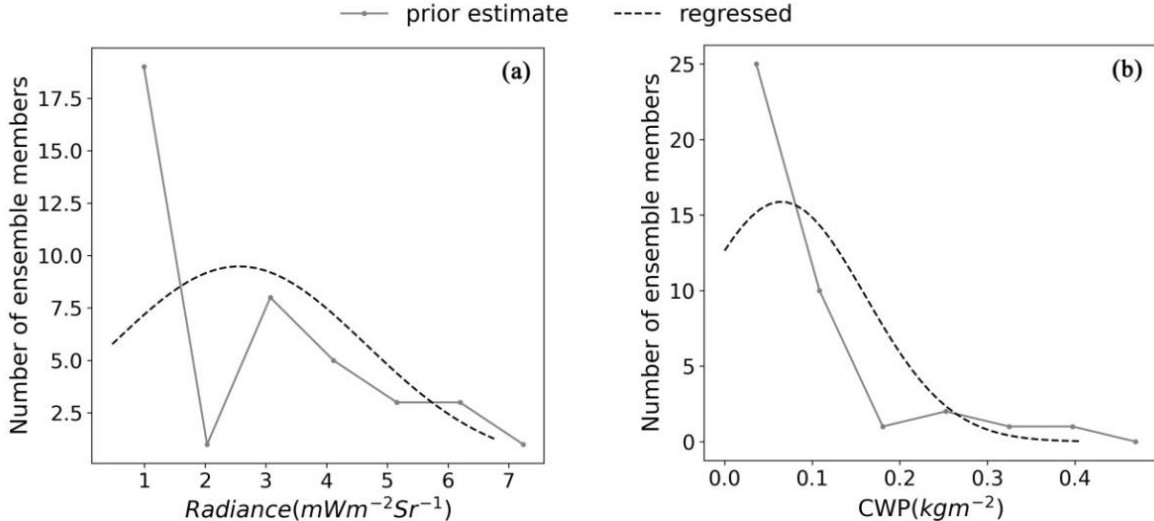

**Figure 6. The first-guess Probability Density Functions (PDFs) of (a) equivalent visible radiance and (b) the cloud water path (CWP). The PDFs were estimated from the 40 prior ensembles for Point 3 (red dot) in Figure 1.**

## 3.2 Cycled DA experiments

The results in this section correspond to the OSSE setups described in Section 2.3.3. The main focuses are the impact of the VIS radiance DA on the analysis and first-guess forecast of CWP, cloud coverage, non-cloud state variables, and precipitation.

### 3.2.1 Impact on CWP and cloud coverage

The time evolution of CWP for the nature run, control run, and the first-guess forecast and the analysis of CWP for Exp-01 are presented in Figure 7.

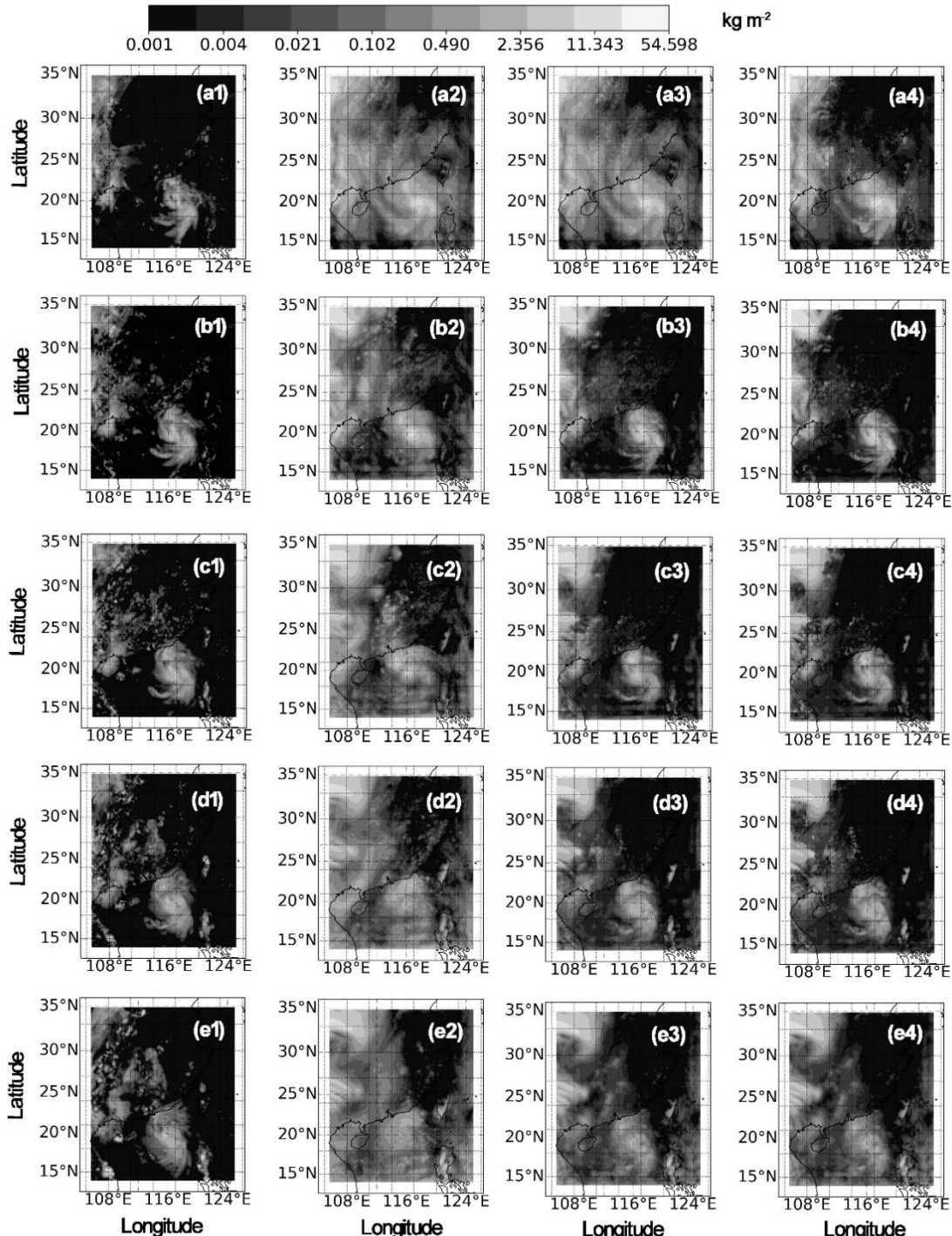

**Figure 7. The time evolution of cloud water path (CWP) for the nature run (column 1), control run (column 2), first-guess forecast (column 3), and analysis (column 4) of Exp-01. From top to bottom, the row panels correspond to 02:00, 04:00, 06:00, 08:00, and 10:00 UTC on 18 August 2020.**

The results indicate distinct differences between the first guess and the analysis of CWP on 02:00 UTC, 18 August 2020 (Figure 7(a3)-7(a4)). After assimilating the VIS radiance data, the horizontal distribution of the ensemble mean first-guess CWP is quite similar to that of the analysis. An extremely large analysis increment of CWP was rarely expected as mentioned in Section 2.4. The resemblance between the cycled DA experiments and the nature run also indicates the improvements on the analysis and the first-guess forecast of CWP and cloud coverage. Compared with the control run, assimilating the VIS radiance data clearly suppressed the false alarm clouds. However, DA of VIS radiance could not generate clouds which were under-predicted. The inability to correct the underprediction was illustrated with a cross-section analysis shown in Figure 8.

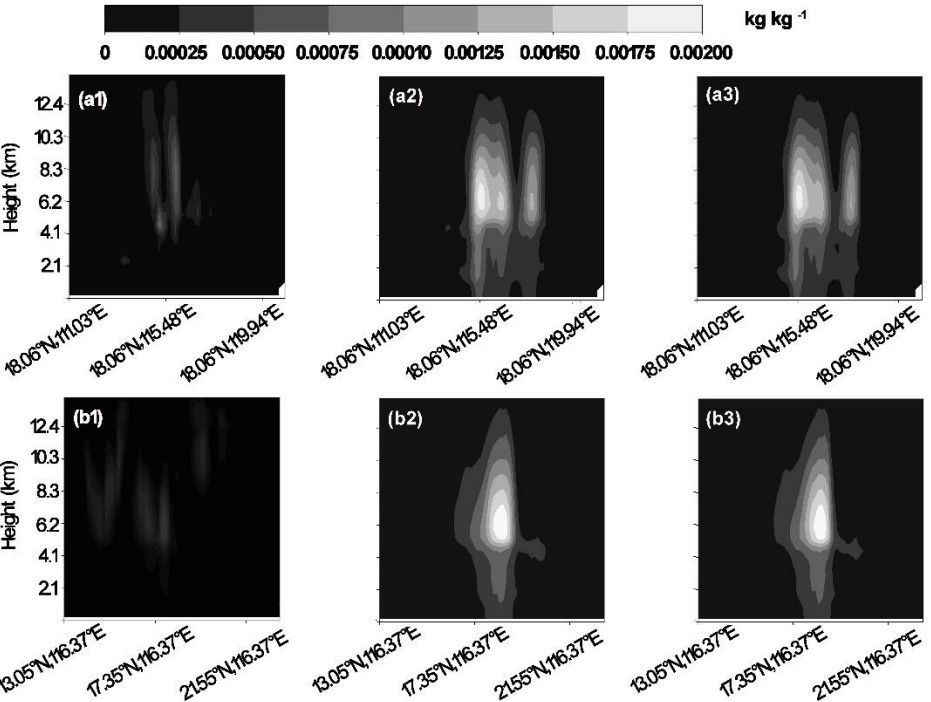

**Figure 8. The x-press cross section of the cloud water and ice mixing ratios of Exp-01 at 02:00 UTC, 18 August 2020 for the AB (the upper panel) and CD (the lower panel) lines shown in Figure 1. From left to right, the column panels correspond to the nature run, the ensemble mean of the first guess, and the ensemble mean of the analysis, respectively.**

Assimilating VIS radiance data did not improve the underprediction of the cloud distributions in vertical and horizontal directions. In the vertical direction, a single layer cloud was presented between 4 and 12 km height in the nature run (Figure 8(a1) and 8(b1)). However, clouds were presented between 0 and12 km height in the first guess (Figure 8(a2) and 8(b2)). After the DA, $Q_c/Q_i$ was decreased but the clouds below 4 km height were not removed (Figure 8(a3) ~ 8(b3)). In the horizontal direction, a thin-layer cloud was presented between 14 °N ~ 16 °N in the nature run (Figure 8 (b1)). This cloud fragment was not simulated by the first-guess atmosphere state (Figure 8 (b2)), nor was it regenerated after assimilating the VIS radiance data (Figure 8(b3)). Therefore, assimilating the VIS radiance would not generate cloud hydrometers for the

region with clear sky in the first guess because there is only zero spread of cloud variables from the unrepresentative background error covariance.

      Quantitative analyses of CWP and CFC indicated improved analysis and first-guess forecasts for the cycled DA experiments. The results are varying with filter algorithms, cycling variables, outlier thresholds, observation errors, and observations with or without thinning, which were discussed in the following.

*a. Influences of filter algorithms*

      To evaluate the performances of different filters on the analysis and first-guess forecast of cloud variables, quantitative analyses of FSS and MAE of the ensemble mean CWP for Exp-01 ~ Exp-03 (EAKF) and Exp-04 ~ Exp-06 (RHF) are presented in Figure 9. In general, the performance of RHF is comparable to or slightly better than EAKF. At some analysis times before 03:00 UTC, the FSS of CWP for the analysis is larger for Exp-01 than for Exp-04, but the first-guess FSS at the

next analysis time is larger for Exp-04 than Exp-01. Similar results were found for the 1 hr and 3 hr first-guess forecasts. These results imply that better analyses do not always ensure better forecasts.

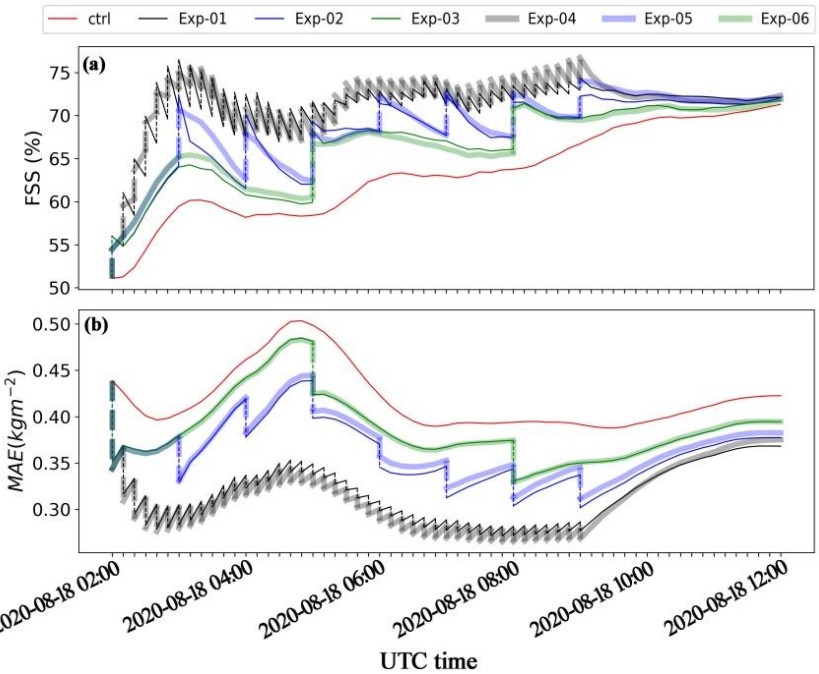

**Figure 9. The time evolution of (a) FSS and (b) MAE of the ensemble mean of the first-guess forecast and analysis of CWP for the cycled DA experiments which are designed to illustrate the sensitivity to filter algorithms.**

Unlike EAKF that assumes a Gaussian prior PDF, RHF does not. However, the performance of RHF is also subject to the sampling errors due to limited ensemble members and other factors as indicated by Anderson (2010). Therefore, only comparable or slightly better analysis and first-guess estimates were revealed for RHF than EAKF. In addition, updating the state variables by RHF is at a sacrifice of more computational cost. For the DA of 20567 observations in one DA cycle at a

Linux cluster equipped with a 2.20 GHz Xeon Silver 4214 CPU with 12 cores, the elapsed CPU time is 775 s and 440 s for
the RHF and EAKF methods, respectively.

*b. Influences of cycling variables*

The single observation experiments indicate that assimilating VIS radiance data only generated trivial impact on non-
cloud variables at 02:00 UTC, 18 August 2020. However, the non-cloud variables are an important part of a cycling DA
system. To explore the impact of including or excluding the updated non-cloud parameters in the ensemble cycles, FSS and
MAE of the 10 min and 1 hr first-guess forecast and analysis of the ensemble mean CWP for the Exp-01 ~ Exp-02 and Exp-
07 ~ Exp-08 groups are analysed. Figure 10 shows that Exp-01 (and Exp-02) outperforms Exp-07 (and Exp-08), indicating
that including the cloud and non-cloud variables in the ensemble cycling makes the forecasting more skilful than cycling the
cloud variables alone. The benefit was introduced to the non-cloud variables during the model integration.

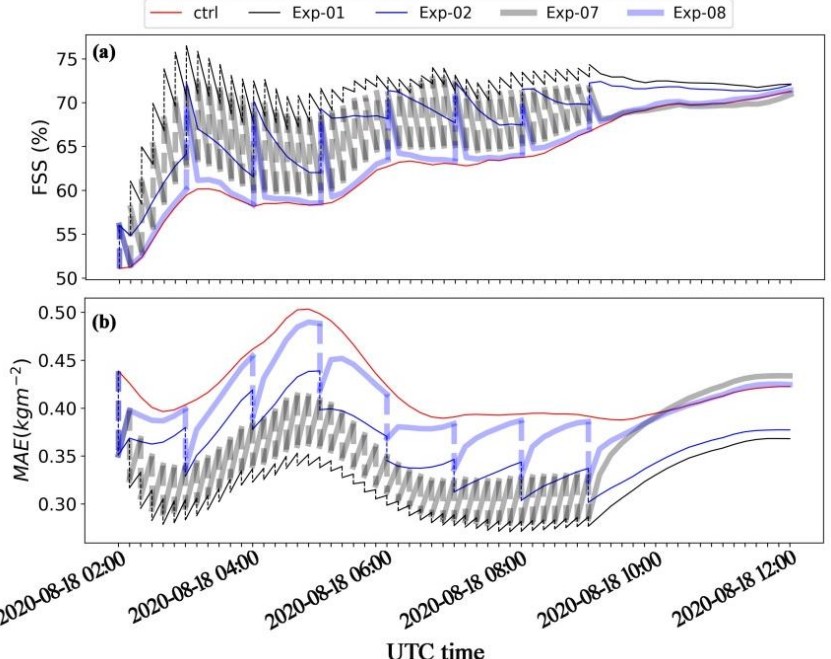

Figure 10. The time evolution of (a) FSS and (b) MAE of the ensemble mean of the first-guess forecast and analysis of CWP for the
cycled DA experiments designed to illustrate the sensitivity to cycling variables.

To demonstrate the error growth for the non-cloud variables, the temporal evolution of the ratio of benefits ⊐, calculated
by formula (15), is presented by Figure 11.

$$⊐ = N_{bet}/N_{eff} \tag{15}$$

where $N_{bet}$ denotes the number of horizontal grid boxes with negative differences of MPE between the analysis and the first
guess (refer to Section 2.4 and Scheck et al., 2020), and $N_{eff}$ denotes the number of observations effectively assimilated by
the DA system (see the next section).

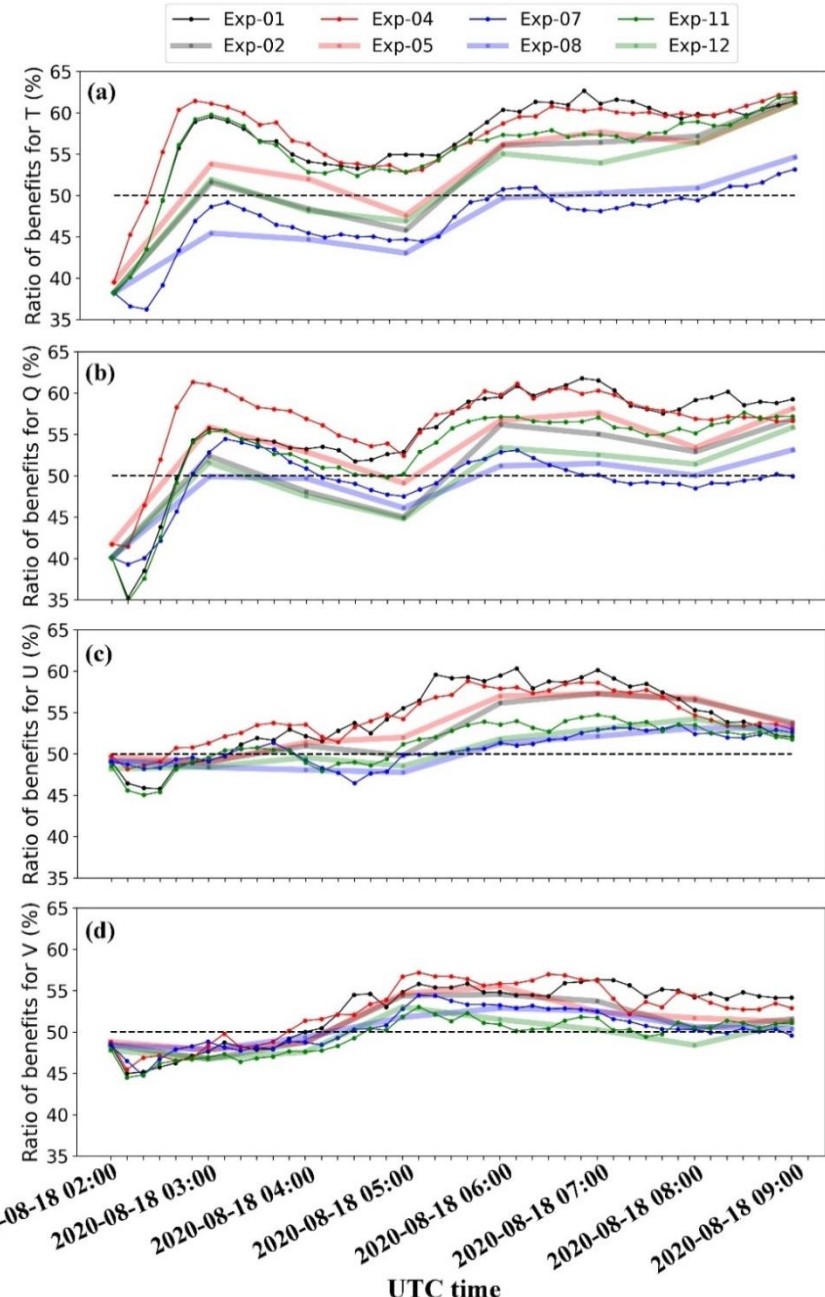

**Figure 11. The time evolution of the ratio of beneficial impact ⊐, calculated by Formula (15) for the first-guess forecast and analysis of (a) the perturbation potential temperature; (b) the water vapour mixing ratio; (c) the x-wind component; and (d) the y-wind component.**

Figure 11 indicates positive impact of SW radiance DA on the major non-cloud variables, especially at the later cycling steps. We think that a main reason is the positive feedback to the non-cloud variables due to the adjustment to cloud variables during the WRF ensemble integration.

A potential explanation to the slightly positive impact of the DA on water vapour mixing ratio is given here. If the ensemble mean of the first-guess equivalent radiance is overestimated for a cloud containing liquid hydrometeors, Qc tend to be decreased in the analysis field in order to match a negative analysis increment in the observation space. In the next ensemble forecast cycle, the grid boxes with decreased Qc may generate more hydrometeors due to condensation. This is because the supersaturation of surrounding atmosphere is increased due to a loss of some cloud hydrometeors (i.e., a loss of

condensation surface). The increased supersaturation could activate new condensation nuclei and enhance the condensation process, which consumes more water vapour. As a result, the VIS radiance could be positively related to the water vapour mixing ratio, and vice versa. According to Formula (4), the covariance between the VIS radiance and water vapour mixing ratio will adjust Q correctly.

    The adjustment of temperature is more likely related to the interactions between clouds and radiation. For example,

decreased CWP or CFC tends to enhance the direct radiation flux in the surface layer, increasing the low-level temperature toward the truth as indicated by Scheck et al. (2020), and vice versa. In addition, the interactions between clouds and longwave radiation tend to generate cooling effects at cloud top and heating effects at cloud bottom (Zhang et al., 2020). Therefore, a relationship between cloud radiance and temperature is expected, and the covariance between VIS radiance and temperature could adjust T correctly.

The impact of SW radiance DA on the x- and y-wind components is slightly negative before 04:00 UTC, and becomes positive thereafter. We think that the positive impact is mainly caused by the convergence and divergence related to the thermal instability, which is closely related to cloud formation (increased radiance) and dissipation (decreased radiance) for the convective systems. As the cloud-radiation interactions modify the temperature profile, the change of clouds could strengthen or weaken the thermal instability and impact the z-wind component. The z-wind component is closely related to

horizontal x- and y-wind components by adjusting the convergence and divergence (White et al., 2018). Therefore, an indirect "radiance-cloud-vertical velocity-convergence and divergence-horizontal wind" interaction could map the observation increment to the U and V increments.

*c. Influences of outlier threshold values*

    Not all observations were effectively assimilated by the WRF/DART-RTTOV system. Some of the observations were

rejected by the DA system due to two reasons: 1) existence of non-monotonic pressures, i.e., pressure increases with altitude. This situation was generated at some points by the improper interpolation of the perturbed first-guess model state to the RTTOV pre-defined layers. For this case study, non-monotonic pressures were mainly located in the Qinghai-Tibet Plateau, Tianshan Mountain, and Central Taiwan ranges (not shown for simplicity), where complex high terrain exists; 2) the differences between the observations and the first-guess ensemble mean equivalent observations are too large and

assimilating these data may cause collapse of WRF model. For the observations rejected due to the second reason, the ratio of observations to be assimilated is mainly determined by setting an outlier threshold value. Increasing the outlier threshold value could increase the ratio of observations effectively assimilated (Figure 12), but it may introduce unstable adjustments to model state variables and destroy the forecast. Therefore, a trade-off between large outlier threshold value and the

potentially detrimental effects on forecasts should be assessed. The analysis and 10 min, 1 hr, and 3 hr first-guess forecasts

indicate improved results when increasing the outlier threshold values (Figure 13) and an outlier threshold value of 6 does not cause the collapse of WRF and generates improvements to the analysis and first-guess forecasts of CWP and cloud coverage.

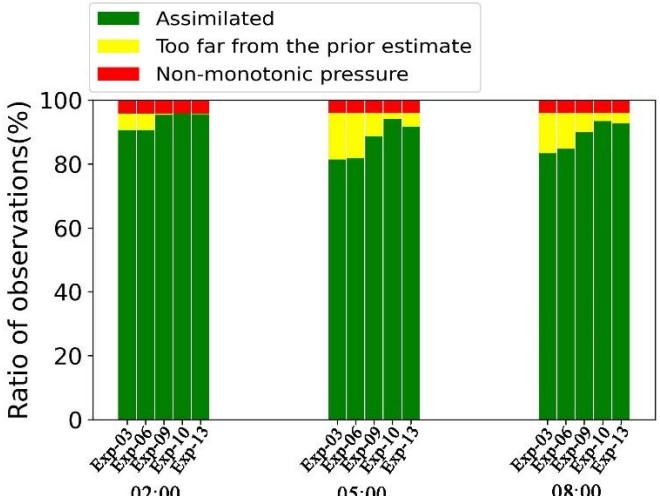

**Figure 12. The time evolution of the ratio of observations that are assimilated or rejected by the WRF-DART system for different**
**experiment designs with different outlier threshold values and observation errors.**

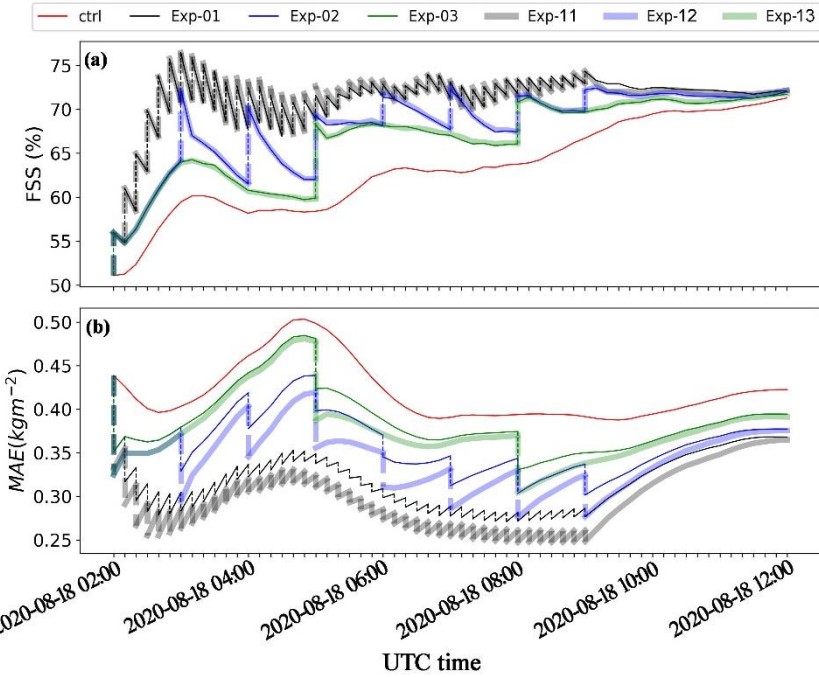

**Figure 13. The time evolution of (a) FSS and (b) MAE of the ensemble mean of the first-guess forecast and analysis of CWP for cycled DA experiments with different outlier threshold values.**

The ratio of observations effectively assimilated is also influenced by the observation errors. Enlarging the observation errors will retain larger ratio of observations effectively assimilated (Figure 12). However, increasing observation errors implies less weight assigned to the observations during the DA. Therefore, the analysis and first-guess forecast are influenced by the observation errors in both ways with a trade-off. To evaluate the influences, FSS and MAE of the ensemble mean CWP for the cycled DA experiments with different observation errors are presented in Figure 14. The

analysis and first-guess forecasts of CWP and cloud coverage are negatively related to the observation errors. As a complementary to the parameters controlling the number of observations, the influences by thinned observations (Exp-14) were presented. The results indicate that thinning observation only resulted in slight improvements.

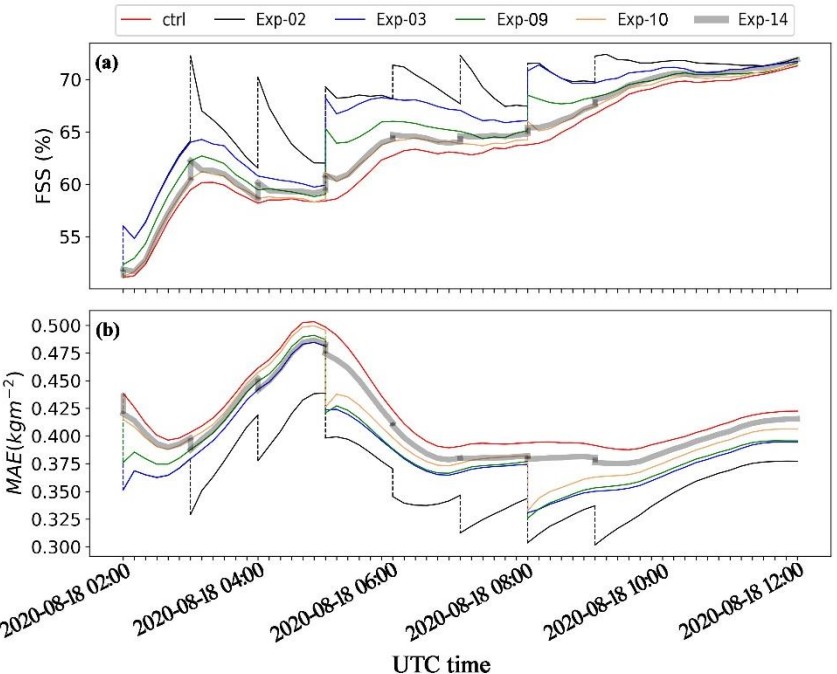

**Figure 14. The time evolution of (a) FSS and (b) MAE of the ensemble mean of the first-guess forecast and analysis of CWP for the**
**cycled DA experiments with varying observation errors and observation pre-processing (with and without thinning).**

### 3.2.2 Impact on precipitation

       The first-guess forecast of rain rate for the nature run, control run, Exp-01, and Exp-11 are shown in Figure 15. On the domain average, the rain rates were overestimated by both control and cycled DA experiments. Compared with the control run, the precipitation for the cycled DA experiments was decreased in most cases, and the spatial distribution of the

precipitation agreed better with the nature run (Figure 16(a)).

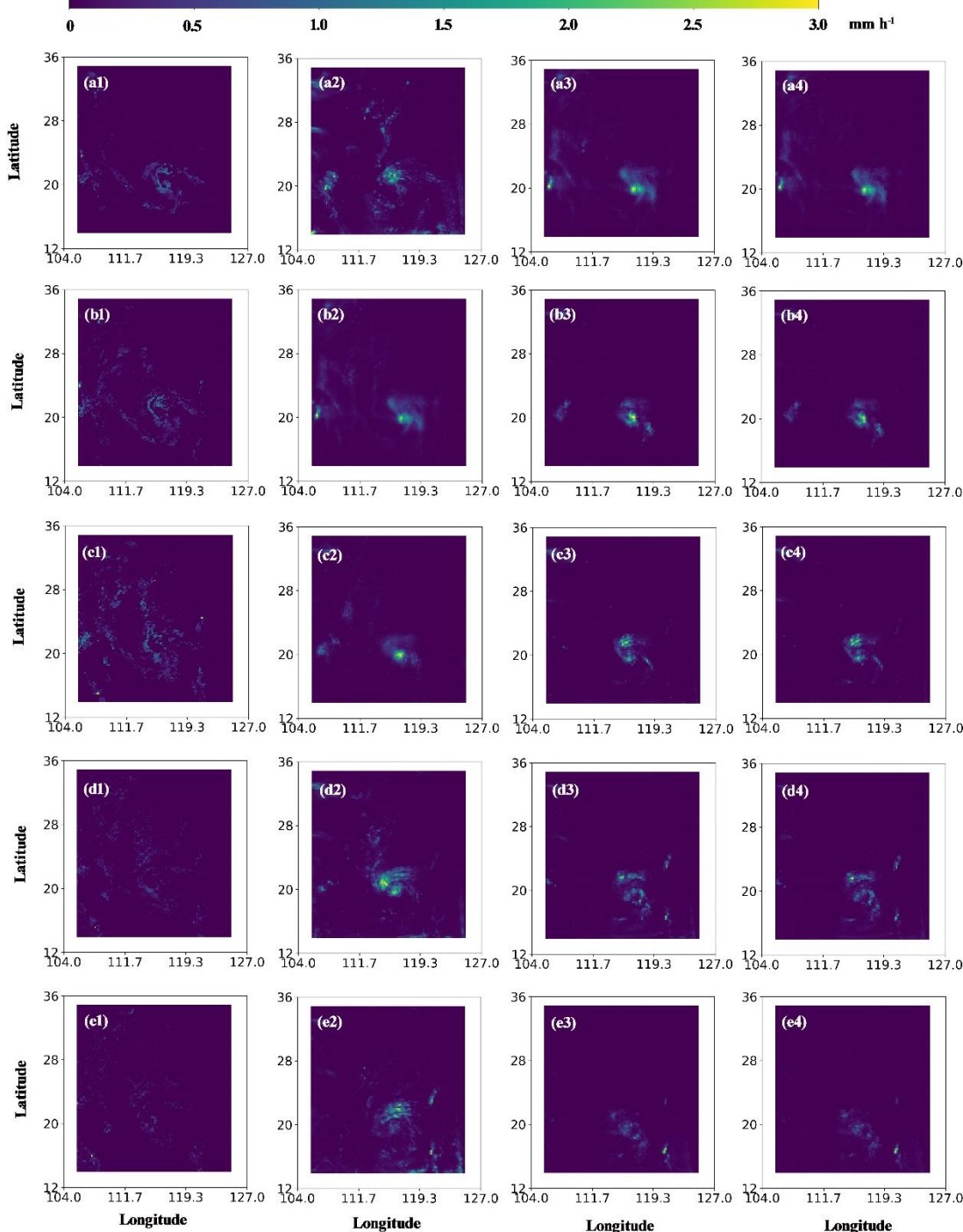

**Figure 15. The time evolution of the rain rate for the nature run (column 1), control run (column 2), first-guess forecast of Exp-01 (column 3) and Exp-11(column 4). From top to bottom, the row panels correspond to the results for 02:00~02:10, 04:00~04:10, 06:00~06:10, 08:00~08:10, and 10:00~10:10 UTC 18 August 2020.**

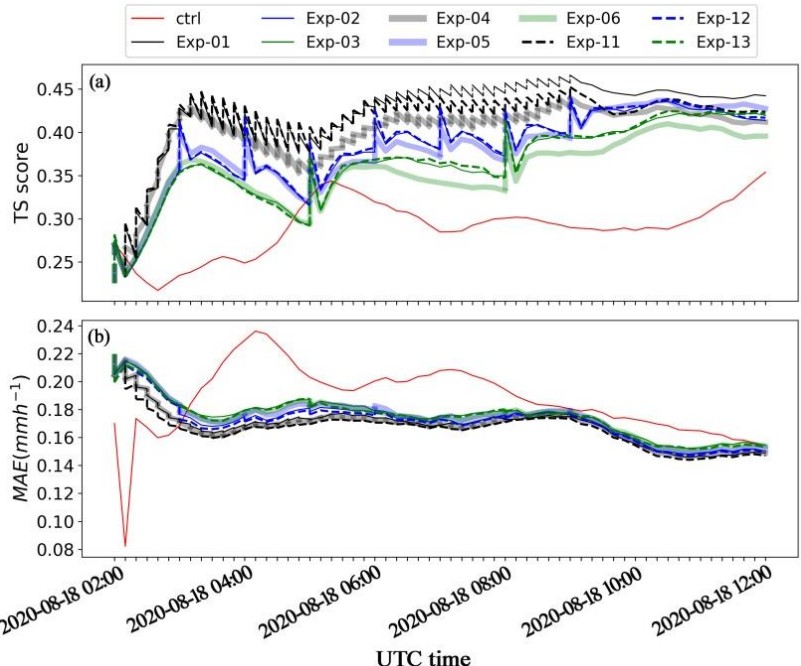

**Figure 16. The time evolution of (a) the TS score and (b) MAE of the ensemble mean of the first-guess forecast of rain rate.**

The metrics of the rain rate forecast indicate that the forecasting skills were improved at most analysis times (Figure 16(b)). However, the improvements on rain rate did not happen at all times. For example, at the initial cycling step (before 04:00 UTC, August 18, 2020), the control run seemed to outperform the cycled DA experiments. As the time progressed, the advantages of assimilating VIS radiance data became apparent. The improved TS score was closely related to the improved CWP and cloud coverage. It is noted that the improvements on the rain rate were not as significant as CWP. Some studies indicated that precipitation was closely related to cloud vertical structure (Kubar and Hartmann, 2008; Yan et al., 2019), the presence and distribution of liquid and ice hydrometeors (Field and Heymsfield, 2015; Mülmenstädt et al., 2015; Korolev et al., 2017), surrounding atmosphere and dynamic state variables (Kanji et al., 2017), etc. Therefore, the impact of the SW radiance DA on the rain rate suffers from its inability to constrain cloud vertical structures, improve cloud phase simulations, and correct cloud location errors (underestimation), etc., as discussed in the previous sections.

## 4. Conclusions and future works

In this study, a series of single observation experiments and cycled DA experiments were performed in an OSSE framework to evaluate the WRF/DART-RTTOV system for assimilating FY-4A/AGRI VIS (channel 2) radiance data. The single observation experiments were designed to reveal the positive impact and limitations of assimilating satellite VIS radiance data on the cloud variables (liquid/ice water mixing ratio and CWP, effective radius of liquid water droplets, and cloud

fraction) and non-cloud variables (water vapour mixing ratio, perturbation potential temperature, and wind). The cycled DA experiments were designed to explore the impact of assimilating VIS radiance data on the analysis and first-guess forecasts

of a tropical storm case with different model settings and observation pre-processing, including varying the filter algorithms, cycling variables, updating frequencies, outlier thresholds, observation errors, and observation thinning. The main findings are as follows.

(1) Single observation experiments exhibit that assimilating the satellite VIS radiance data generated overall positive impact on cloud variable analysis for the first DA cycle although in some rare cases, the DA increased the errors both in the

575 observation space and in the cloud variable (or diagnosed parameter CWP) space. The negative impact was closely related to the non-linear properties of the forward operator and the non-Gaussian properties of the first-guess probability distribution function. In addition, neutral impact was revealed on non-cloud variables including water vapour mixing ratio, temperature, and horizontal winds.

(2) Although neutral impact was revealed for the non-cloud variables in the first DA cycle, both non-cloud variables

and cloud variables were improved in the following-on ensemble forecast and DA cycles. The beneficial impact on the non-cloud variables was closely related to the analysis increments of cloud variables by condensation/evaporation and deposition/sublimation processes, and cloud-radiation interactions of the model microphysics. The RHF filter slightly outperformed the EAKF filter due to its ability to deal with the non-Gaussian problems, nevertheless at a cost of ~1.8 times of more computational time.

(3) The cycled DA experiments revealed that the first-guess forecast and analysis are positively related to increasing the outlier threshold value and the updating frequency, and negatively related to increasing observation errors and thinning length scale. Similar results were found for the precipitation forecasts. Assimilating the SW radiance data effectively improved the representation of locations with or without precipitation, but less for the quantitative metrics of the rain rate. The limited impact on the rain rate is due to the fact that the DA scheme is not effective to constrain cloud vertical structures,

modifying cloud phases, and correcting cloud location errors in the case of underprediction that caused zero-spread problem.

The findings in this study provide useful guidance on setting WRF/DART-RTTOV configurations and pre-processing of observations for assimilating the FY-4A and the upcoming FY-4B VIS radiance data in real-time models. This study complements to Scheck et al. (2020) who reported the potentials and limitations of assimilating the Meteosat SEVERI VIS imagery with the COSMO/KENDA system using LETKF. In general, the pros and cons of assimilating the VIS radiance

data found in this study agrees to those in Scheck et al. (2020), except that slightly positive impact on horizontal wind speeds was found in this study while Scheck et al. (2020) reported neutral impact. We tentatively ascribe the positive impact on U and V to the convergence or divergence related to the "radiance——cloud——vertical velocity——convergence and divergence——horizontal winds" interaction and it should differ in weather systems. Therefore, the different impact on horizontal winds of the two studies could be caused by the differences in the weather systems studied. In addition, the two

studies used different the models/tools and corresponding configurations and this study explored the properties unique to the WRF/DART system.

It is noteworthy that the present study was based on the low-resolution model simulations at 15 km × 15 km. Such grid spacing is large enough to avoid radiance simulation errors due to 3D radiative effects, which are apparent for high-resolution simulations (Várnai and Marshak, 2001). Although the 3D radiative transfer effects could be properly corrected by some of the forward operators in RTTOV, the related parameters and datasets specific to FY-4A/AGRI are currently unavailable. Further studies should be extended to cloud-resolving model simulations like Scheck et al. (2020) to fully take the advantage of high-resolution satellite VIS radiance data. Meanwhile, some attention should be paid to the forward operator because the enhanced 3D radiative effects in high-resolution modelling may cause the nonlinearity of forward operator more complicated. An outlook of future works is summarized below.

(1) To tune up the forward operators and more accurately estimate the observation errors. The findings in this study suggest that decreasing observation errors improves the first-guess forecast and analysis. One of the factors determining the observation errors is the forward operator (Janjić et al., 2017). Therefore, it is necessary to tune up the RTTOV configurations in simulating synthetic FY-4A visible imagery from WRF model state variables and to estimate the observation errors under different modelling resolutions, weather conditions, sun-viewing geometries, etc. Scheck et al. (2018) assessed the performance of the forward operator MFASIS by comparing the synthetic visible imagery simulated based on the state variables from COSMO with the SEVIRI visible image. These works could be referenced when assessing the performance of RTTOV simulations against FY-4A visible observations.

(2) To improve the computational efficiency and accuracy of the forward operators. Assimilating visible radiance data is quite time-consuming for the current WRF/DART-RTTOV system (around 7 min in an EAKF cycle and 13 min in a RHF cycle). Increasing the updating frequency and outlier threshold value makes the computational cost even more expensive. We need an accurate and fast observation operator for assimilating the FY-4A (and the upcoming FY-4B) visible radiance data at both low- and high-resolution simulations. Scheck et al. (2016a) and Scheck (2021) developed a Look-up Table (LUT) and machine learning based forward operator that is several orders faster than DOM-based methods. In addition, three-dimensional radiative effects could be corrected for high-resolution modelling without adding too expensive computation cost (Scheck et al., 2018; Albers et al., 2020; Zhou et al., 2021). These methods should be explored to improve forward operators suitable for assimilating the FY-4A visible radiance data.

(3) To correct the errors due to the non-Gaussian and non-linear problems with the SW radiance DA. The performance of EAKF was limited in dealing with the non-linear and non-Gaussian problems. The Particle Filter (PF) is declared to have advantages in dealing with the non-Gaussian and non-linear problems. With certain localized method included, PF shows great potential in application to high-dimension numerical prediction model such as the WRF model (Shen and Tang, 2015; Poterjoy, 2016; Pinheiro et al., 2019). Therefore, some newly-developed PF methods could be a candidate to further improve the forecasting skills of WRF model with the DA of satellite visible radiance.

(4) To develop techniques to reduce the cloud location errors. The performance of WRF/DART system is limited to correct the cloud location errors in the case that the first guess underpredicts clouds or a zero spread of prior ensemble members occurs. Dowell et al. (2011) proposed a method to tackle the location errors in assimilating radar data by EnKF.

Their basic idea was to add some perturbations to the base state randomly and add local perturbations in and near precipitation areas regularly to produce clouds in the precipitation areas. In addition, White et al. (2018) also promoted a method to produce clouds comparable to satellite observations, but this method needs additional observations (such as brightness temperature at infrared bands) to support. Improvement on horizontal cloud location errors could improve precipitation forecasts in some cases. For example, the vertical cloud structure is less important for some deep convection since the cloud may extend from boundary layer up to troposphere (Hu et al., 2021b).

(5) To evaluate the impact of assimilating FY-4A visible radiance data for long-term forecasts. The limited impact of the DA on rain rate is partly caused by the shortcomings of the DA procedure discussed above, but by the spin-up effects as well. The spin-up effects may introduce false alarm precipitation due to the interactions between the model dynamics and microphysics when smaller scales are now well represented in the initial conditions and lateral boundary conditions (Short and Petch, 2022). In this study, we started to assimilate the synthetic FY-4A visible radiance data after 2-hour cold start and run the model for 10-hour forecast (02:00 ~ 12:00 UTC). This is a too short time period to exclude the spin up. Improvements on precipitation should be expected for longer forecasts.

Finally, for the real FY-4A visible radiance DA, the NWP model errors on cloud and precipitation forecasts should be considered in the DA processes. On one hand, the parameterization (such as the subgrid-scale cloud fraction parameterization in this study) is closely related to the calculation of synthetic visible radiance by a forward operator. On the other hand, the formation and dissipation of cloud and precipitation highly depend on the model parameterization. Suboptimal parameterization may introduce large model bias in some cases due to unsolved scales and processes (Janjić et al., 2017). The model bias could introduce negative impact on cloud and precipitation forecasts. Therefore, the NWP model should be tuned to properly represent the scale-dependent microphysical processes in order to fully realize the effects of the FY-4A visible radiance DA.

**Code and data availability**

Version 4.1.1 of WRF-ARW source code is publicly available at http://www2.mmm.ucar.edu/wrf/users/. The Manhattan release of DART source code (version 9.8.0), including the RTTOV observation operator (version 12.3), is publicly available at https://dart.ucar.edu/. Version 12.3 of RTTOV source code is publicly available at https://nwp-saf.eumetsat.int/site/software/rttov/. The NCEP FNL (Final) Operational Global Analysis data are downloaded from https://rda.ucar.edu/datasets/ds083.2/. The ERA5 hourly data are available at https://cds.climate.copernicus.eu/api/v2/resources. The source codes of WRF-ARW, WPS, RTTOV, and DART models (tool), as well as the input and (processed) output data, and the visualization scripts are available at https://doi.org/10.5281/zenodo.7028828 (Zhou et al., 2022).

**Author contribution**

Yongbo Zhou: Conceptualization, Methodology, Writing original draft, Visualization, Funding acquisition. Yubao Liu: Conceptualization, review and editing, Funding acquisition. Zhaoyang Huo, review and editing. Yang Li, review and editing.

**Competing interests**

The authors declare that they have no conflict of interest.

**Acknowledgement**

This study is supported by the Natural Science Foundation of Jiangsu Province (Grant No. BK0210665) and Startup Foundation for Introducing Talent of Nanjing University of Information Science and Technology (Grant No. 2019r095). The work is also partly supported by the National Key R&D Program of China (Grant No. 2018YFC1507901). We are grateful to

Glen Romine, Moha Gharamti, Nancy Collins, and Tim Hoar from the DART team at the National Center for Atmospheric Research (NCAR) for kindly providing instructions on running this tool. We acknowledge the High Performance Computing Center of Nanjing University of Information Science & Technology for their support of this work. We are grateful to the two anonymous reviewers for their constructive comments and suggestions.

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
