# Peer review of "A preliminary evaluation of FY-4A visible radiance data assimilation by the WRF (ARW v4.1.1)/DART (Manhattan release v9.8.0)-RTTOV (v12.3) system for a tropical storm case"

_Geoscientific Model Development, 2022_

## Author Comment (AC1)

**Response to reviewer #1**

**Aim and relevance of the paper, Title and abstract**

The present paper evaluates Observing System Simulation Experiments of a future satellite with the WRF-DART system. Radiances in the visible range are assimilated for a cyclone case. The authors demonstrate an improvement of the forecasts of cloud-related parameters and reveal weaknesses of the method.

Visible range satellite radiance assimilation is a rather recent field of research and evaluating the impact of a new satellite in OSSEs for a critical weather event is a future-oriented approach. Therefore, the present paper is highly relevant for the community. It promotes research in multiple fields at the same time: visible range radiance assimilation, the exploitation of a new satellite, and research on cyclones.

The title is informative and contains all relevant information, except maybe for the name of the satellite. Adding the name of the satellite to the title would make the word "preliminary" more meaningful.

Throughout the whole paper, the satellite is mentioned as "FY-4". However, FY-4 is a series of satellites. I believe the study uses FY-4B and this should be clearly mentioned throughout the paper.

**Our Response:**

This study focuses on the data assimilation of FY-4A satellite visible radiances. The name of FY-4A satellite was added to the title of the revised manuscript. After correction, the title of this manuscript is "A preliminary evaluation of FY-4A visible radiance data assimilation by the WRF/DART-RTTOV system for a tropical storm case". Since launch in 2016, FY-4A satellite provides abundant visible radiance data. However, these data are not currently assimilated in any of the operational numerical whether prediction centers. In addition, RTTOV is a new capability of WRF/DART system (quoted from the DART website *https://docs.dart.ucar.edu/en/latest/guide/Radiance_support.html?highlight=RTTOV*). Therefore, we added the word "preliminary" to the title. (L1-2)

Accordingly, the satellite name "FY-4" was replaced by FY-4A throughout the revised manuscript.

The abstract misses two important pieces of information:
1)      For which cyclone case was the study performed? Why was a cyclone case, and more specifically "this" cyclone case chosen for this pilot study of FY-4 SW radiance assimlation?

**Our Response:**

We focus on the pre-landfall stage of a cyclone case, which is a tropical storm system named Higos. The tropical storm case occurred over the West Pacific in 2020. We choose this tropical storm case for the following four reasons. (L132-139, L163)

1) The cloud distribution of a tropical storm has a very typical anti-clockwise structure in the horizontal direction, which is intuitively clear to assess the forecasting skills of cycled data assimilation experiments. Therefore, we choose the tropical storm case to perform the numerical simulations.

2) The tropical storm case was chosen because abundant FY-4A visible observations were only available at daytime. During this period of time, FY-4A visible radiances could capture the development of the tropical storm. Some tropical storm cases were accompanied with very large solar zenith angles or beyond the observation range of FY-4A. These tropical storm cases were not suitable for this study.

3) The tropical storm case consists of multi-layer and mixed-phase cloud structures and precipitation properties, which facilitates the demonstration of the beneficial impacts on the analysis and first-guess forecast of CWP, cloud coverage, and precipitation. It is also designed to reveal the inabilities and ambiguities of the assimilation to improve cloud vertical structures and phases, especially for the multi-layer and mixed-phase cloud structures.

4) After a 14-hr spin up time, the nature run (theoretical true model state) could capture the bulk track properties which agree well with the observations, which makes the results relevant to this tropical storm more robust.

2) At the end of the abstract, an outlook is missing. What do the results imply? What should be future steps of research?

**Our Response:**
The main findings of this study are that assimilating the visible radiances could clearly improves the analysis and first-guess forecast of cloud water path and cloud coverage. However, the assimilation cannot constrain cloud location errors especially for under-predicted clouds. Sometimes, the non-linear and non-Gaussian problems could bring negative impacts. In addition, the elapsed CPU time for a data assimilation cycle is about 7 min for EAKF and 13 min for RHF, which is too slow for operational application. As a result, only slightly positive impacts were achieved on the analysis and first-guess forecast of precipitation because the precipitation is closely related to cloud vertical strictures and cloud phases, and et al. (L17-26)

The results imply that future works should focus on the development of faster and accurate forward operators suitable for the assimilation of FY-4A visible imagery, on the techniques to reduce the non-linear and non-Gaussian errors, and on the methods to correct the location errors, etc. Such outlook was added to the abstract of the revised manuscript. (L26-28)

The fact that different parameter settings were tested is important and should be mentioned in the abstract. This is clearly a strength of this paper.

**Our Response:**

Relevant sentences were added to the abstract of the revised manuscript. For the experiment designs, we added "Single observation experiments and cycled DA experiments were performed to explore the pros and cons and sensitivities of the assimilation to different filter algorithms, cycling variables, outlier threshold values, and observation errors." (L15-16)

For the results concerning to the experiment designs, we added "WRF could capture better CWP and CFC properties for the Rank Histogram Filter (RHF) than the Ensemble adjustment Kalman Filter (EAKF) but with a sacrifice of more elapsed CPU time ($\approx$ 7 min for EAKF and 13 min for RHF in one cycle), for the cycling variables including both cloud and non-cloud variables, for the larger outlier threshold values, and for the smaller observation errors without thinning of observations". (L18-21)

**Other remarks concerning the abstract:**

L15: You might want to state that FY-4(B?) is a geostationary satellite located over Asia.

**Our Response:**
In fact, we choose FY-4A satellite to perform the numerical simulations. Since launch in 2016, FY-4A satellite provides abundant visible radiance observations. However, these data are not currently assimilated in any of the operational numerical weather prediction centres. Therefore, we think a simulation study based on FY-4A satellite is meaningful. Because the designs of the AGRI payload for the two satellites are rather similar, we believe that the results should be representative to the upcoming FY-4B VIS radiance data. (L12, 27)

L16: You mention the experiment for which the best results were obtained without explaining what kind of experiments have been performed.

**Our Response:**
Single observation experiments and cycled DA experiments were performed to explore the pros and cons and sensitivities of the assimilation to different filter algorithms, cycling variables, outlier threshold values, and observation errors. (L15-16)

L18: As the previous sentence already contains "best resutls", I suggest to modify the beginning of this sentence to for example: "In this case, WRF could capture [...]"

**Our Response:**
Corrected. (L18)

L18: I suggest to modify the end of this sentence like this: "[...] and significantly improve the cloud water path and cloud coverage forecast."

**Our Response:**
Corrected. (L17-18)

L19: What does the word "its" refer to here? The simulation system? In this case you might write "The first is that the simulation system..."

**Our Response:**
Its refers to assimilating the VIS radiance data by the WRF/DART-RTTOV system. For simplicity, we use the word "the assimilation". (L24)

**Specific comments and remarks**

**#Introduction and background:**

L24: I suggest to add the word "satellite" to the beginning of the second sentence: "Most satellite DA-related studies [...]".

**Our Response:**
Corrected. (L32)

L33: "[...] only provide information on cloud top microphysics [...]"

**Our Response:**
Corrected. (L53)

L33: Better replace "weather radar" by "precipitation radar"

**Our Response:**
Corrected. (L54)

L37: This sentence is a bit misleading, you might want to say it like this: "Therefore, high-resolution satellite SW radiances provide information on cloud properties with a great significance for cloud-resolving model simulations."

**Our Response:**
Corrected. (L57-58)

L52: I suggest to change "in assimilating satellite radiance data" to "in satellite radiance DA".

**Our Response:**
Corrected. (L77)

L58: I suggest to remove the word "Nowadays".

**Our Response:**

Corrected. (L83)

L74: In my opinion there is no need to put the word hybrid in double quotes. Also, if you mention that "great achievements" have been made, you should state what these achievements are.

**Our Response:**

The double quotes are deleted in the revised manuscript. (L100)

Some details on the achievements of the hybrid DA methods were added to the revised manuscript. (L101-108)

Bauhner et al. (2013) evaluated an ensemble-variational DA approach in assimilating the observations which are operationally assimilated in Environment Canada, and concluded that the hybrid method is more skilful than variational methods to improve the short- and middle-range forecasting over tropical and extra-tropical regions. Gao et al. (2013) developed a hybrid Ensemble Kalman Filter (EnKF)-3DVAR method to effectively assimilate radar data. The hybrid EnKF-3DVAR method outperforms 3DVAR or EnKF in shortening the spin-up time of a supercell storm. In addition, the hybrid method is increasingly applied in satellite radiance DA. Xu et al. (2016) assimilated the FY-3B satellite MV radiance data by the WRF hybrid ensemble/3DVAR method, better forecasts of typhoons' track, intensities, and precipitation were reported compared with 3DVAR. Similar results were also reported by Shen et al. (2020).

L80: You mention that RTTOV was "recently" enabled for DART. Do you have a reference for that information? Otherwise the word "recently" does not make sense.

**Our Response:**

The word "recently" was quoted from the DART website https://docs.dart.ucar.edu/en/latest/guide/Radiance_support.html?highlight=RTTOV. On this page, there is a paragraph entitled "Introduction to DART's support for RTTOV", which says that "DART now includes the ability to use the RTTOV forward operators ……. This is a new capability for DART ……". Since this page is edited in 2021, we added the word "Recently" in the original manuscript. In the revised manuscript, we deleted the word "Recently". (L111)

L88: "Section 3" not "Sections 3"

**Our Response:**

Corrected. (L127)

**#References:**

The provided references are relevant and recent and include key studies in the field.

**#Methods:**

You should probably add sources for the FNL and ERA5 data sets.

**Our Response:**
Corrected. (L154, L170)

L91: So far nothing has been demonstrated and this sounds like a sentence from the conclusion. Maybe build this sentence like this: "This study demonstrates the performance of the WRF/DART-RTTOV [...]". Also, add the relevant information from the abstract: Mention FY-4 for example.

**Our Response:**
We hope to evaluate the performance of the WRF/DART-RTTOV system in assimilating the simulated FY-4A VIS radiances in an OSSE framework. Such information was already introduced in the last paragraph of Section 1. Therefore, no more tautology is given here. (L119-120)

L99: Better: "horizontal grid boxes" instead of "horizontal grids".

**Our Response:**
Corrected. (L143)

L114: Why did you deviate from the CONUS physics suite for the microphysics scheme?

**Our Response:**
We made a terrible mistake in the original manuscript by randomly choosing the WRF model configurations. In the revised manuscript, the microphysics schemes were chosen with caution. The WRF model configurations include the Thompson microphysical scheme, the Tiedtke Cumulus Parameterization option, and the UV planetary boundary layer scheme, which are the optimal schemes for typhoon simulations over the Northwest Pacific Ocean as suggested by Di et al (2019). (L154-160)

L115: This is the first time the reader learns about the dates of your experiments.

**Our Response:**
The nature run and the cycled data assimilation experiments have different spin-up times. Both of the simulations ended at 12:00 UTC, 18 August, 2020. The initial times of different simulations are different. We believe the dates of different

simulations were better introduced in the revised manuscript. (L161-163 for the nature run, and L177-180 for the cycled data assimilation experiments)

L119: The Betts-Miller-Janjic cumulus scheme is rarely chosen in the WRF literature. Can you explain why you chose this cumulus scheme?

**Our Response:**

In the revised manuscript, we chose the Tiedtke Cumulus Parameterization option as suggested by Di et al. (2019). (L155)

L131: "It is noted that the Baran-2014 scheme has no explicit dependence on ice particle size." - and probably that is the reason why it was used?

**Our Response:**

Exactly. The cirrus scheme developed by Baren et al. (2014) was used to calculate ice cloud optical properties, which has no explicit dependence on ice particle size. Therefore, the effective radius of ice particles was not analyzed in the following sections. As a result, the analyses of the results were simplified given that cloud variables were adjusted collectively but we do not have to analyze the effective radius of ice particles. (L202-204)

L176: Didn't you already mention that you use 50 ensemble members?

**Our Response:**
Corrected. (L253)

L179: And probably that is the reason why you find that the vertical structure of the clouds is not very well represented in the simulations?

**Our Response:**

That's true. The synthetic VIS radiance observation is not sensitive to cloud vertical structures but to the accumulated cloud water/ice mass. As a result, it is difficult to correct cloud vertical location errors due to a lack of information on cloud top height and vertical cloud extent. In addition, the assimilation cannot remove the false alarm liquid clouds due to the spurious covariance between the VIS radiances and liquid water clouds in the background. (L372-375)

L184: And we do have a non-Gaussion problem here, right? This is why this information is given?

**Our Response:**
The RHF assumes a prior probability density distribution function which is a piecewise constant between two ensemble members. The left tail of the PDF and the right tail of the PDF follows a Gaussian function. In general, the RHF prior PDF does not have to be a Gaussian function. (L256-261)

We do have non-Gaussian prior PDF in our study, an example of the non-Gaussian prior PDF is given by Figure 6. (L417).

L188: Please give some more information about the cyclone event. Did the cyclone have a name? Which was the cyclone category at the time of the experiments? From where to where did it move? Does the type of cyclone event not have any influence on the simulations?

**Our Response:**

The tropical storm case is named Higos, which is initially occurred over the north of Luzon, the Philippines as a tropical disturbance on August 16, 2020. The system tracked northwest toward the South China Sea, and intensified into a tropical storm on 19:00 August 17. The tropical storm was further developed into a typhoon system on 12:00 UTC, August 18. The typhoon system landed on Zhuhai, Guangdong province on 22:00 UTC, August 18, and degenerated into a tropical depression on 12:00 UTC, August 19. This study focuses on the pre-landfall stage of the tropical storm (00:00 UTC ~ 12:00 UTC, August 18) under the consideration that FY-4A visible imagery is only available at daytime. (L132-136)

We think that the type of cyclone event should have non-negligible influences on the simulations. For example, if the tropical storm case contains multi-layer and mixed-phase cloud structures, assimilating the visible radiances could not improve cloud vertical structures and phases. In addition, assimilating the visible radiance data generated slightly positive impacts on horizontal winds in this study, while neutral impacts were reported in Scheck et al. (2020). We ascribe the different impacts to the different characteristics of the weather systems. (L607-612)

**#Results:**

L230: Given how many details you provided in chapter 2, you should explain how DA actually changes the base state.

**Our Response:**

The impacts of assimilating visible radiances on the base state were explored by a set of single observation data assimilation experiments and cycled DA experiments. We chose four basic thermal and dynamic variables for detailed discussions, including water vapor mixing ratio (Q), perturbation potential temperature (T), and - and y-wind components (U and V).

The single observation DA experiments revealed neutral impacts on the state variables at the first analysis time. (L340-355)

The cycled DA experiments revealed slightly positive impacts on the base state. The positive impacts were generated due to the feedbacks of the non-cloud variables to the adjustments of cloud variables. The feedbacks include the condensation, evaporation, freezing, and cloud-radiation interactions. (L482-515)

L232: Stating that something is "rather complicated" is not scientific. Please improve this sentence and explain what was complicated about it.

**Our Response:**

The assimilation was complicated by the non-linear and non-Gaussian problems. This part is corrected. (L334-339)

L280: "As indicated" -> Indicated where?

**Our Response:**

What we mean is "As indicated by Figure 5". In the revised manuscript, we changed this sentence to "To see how well the assumption was respected, $p(x)$ in the CWP space are presented by Figure 6 …". (L407)

L305: This could have been explained above.

**Our Response:**

A set of new experiments were performed, and similar tautology was corrected in the revised manuscript. (L435-436, L448-449)

L359: What is a "weak" cloud? This is not a very scientific term.

**Our Response:**

The "weak" was intended to describe clouds with small CWP. This part was completely changed. This non-scientific term was deleted in the revised manuscript.

L362: Do you have an idea why precipitation was not simulated in any of the DA experiments? It is indeed important to mention that, but you should also try to provide reasons.

**Our Response:**

We made a mistake for not analyzing precipitation in the original manuscript. Precipitation should be included in the study because previous studies have shown that assimilating visible radiance data is beneficial to precipitation simulations. In the revised manuscript, results relevant to precipitation was added (L549-573)

L386: "QC = 7" and "QC = 4" are very DART-specific statements that only very few readers would understand, please rephrase this in an understandable way. This is also valid for Figure 11.

**Our Response:**
Corrected. (L516-535)

L392: What do you mean by "far observations"? What this mentioned before?

**Our Response:**

If the difference between the observations and the prior ensemble mean equivalent observations are so large that the observations were rejected by the data assimilation system, these observations is denoted as "far observations". The phrase is misleading and is deleted in the revised manuscript.

**#Discussion and Conclusions:**

L436: You should explain what exactly is meant by "dense" here.

**Our Response:**

In the original manuscript, the "dense" observation means that no thinning were added or thinning with small thinning length scale were added to the observations. In the revised manuscript, we deleted the word "dense". Instead, we use the phrases "observations without thinning" or "observations with small thinning length scale". (L21, 286, 583, 598)

L445: Is it unable to influence the state variables in all the performed experiments?

**Our Response:**

The influences on state variables were discussed by single observation DA experiments and cycled DA experiments. The single observation DA experiments reveal neutral impacts and the cycled DA experiments revealed slightly positive impacts. The single observation DA experiments were performed at the first analysis time without forecasting. The cycled DA experiments were performed with model forecasting, and positive feedbacks were introduced to non-cloud variables due to condensation, freezing, evaporation, and cloud-radiation interactions.(L340-349, L482-515)

L452: This is a bit short as a final sentence and an outlook is missing. What are the most urgent opportunities for future research? Once real FY-4B data will be available, should another cyclone case be used to validate the results? Please provide some more outlook on such questions.

**Our Response:**

A more detailed outlook of future works was added to the revised manuscript, including the optimization of forward operators and estimation of observation errors, the improvements on computational cost and accuracy of forward operators, the correction of errors due to non-Gaussian and non-linear problems, and techniques to reduce cloud location errors. (L620-652)

The experiment set ups were focused on the tropical storm Higos. Tropical storm is a typical Mesoscale weather system, which consists multi-layer and mixed phase cloud structures, and precipitation processes. We think the results should be representative to a cluster of Mesoscale convective weather systems. However, we are

very cautious to draw the conclusion that no more cyclones are needed to validate the results in this study. Instead, we are planning to assimilate the real FY-4A visible radiance observations in real cases to validate the findings of this study.

**Figures and tables**

Axis labels in the figures should start with capital letters. This should be corrected in Figures 2, 3, 4, 5, 6, 8, 10, 11

**Figure 1:**
- The colorbar misses a label.
- You might want to change the color of the ocean to blue instead of green.

**Our Response:**
Corrected. (L150)

**Figure 2:**
- I would remove the "unit:" in the axis labels unless this is required by the journal.

**Our Response:**
Corrected. (L220)

**Figure 3:**
- The resolution of this figure does not seem to be very good, it is a bit blurry. For example in (a2) it is almost impossible to distinguish the lines "iwc-prior mean" and "iwc-posterior"
- The "x10^-4" in the horizontal axis label is a bit lost in all subplots on the right side. Please improve this.

**Our Response:**
This figure is totally changed. All these problems mentioned above should be corrected. (Figure 4, L370)

**Figure 4:**
- What is the "R statement" on the right in the plot? Must it be there?

**Our Response:**
The R statement is to demonstrate the difference of the theoretical true observation increment ($R_{inc}^t$) and the estimated observation increment ($R_{inc}$). Due to the non-linearity of the forward operator, $R_{inc}^t$ and $R_{inc}$ should be different. The R statement was deleted in the in the revised manuscript (Figure 5, L415)

**Figure 5:**
- It is a good idea to make 1-4 correspond to a-d. But one has to search quite a bit to find the indication of the panels (a), (b), (c), (d). Please make these more visible,

for example place them in the top left corner of each panel and in bold and with a larger font size.

**Our Response:**

Thank you for this suggestion. This Figure is totally changed in the revised manuscript. But similar problems in other figures were corrected. (Figure 6 in the revised manuscript, L417)

**Figure 11:**

- The choice of line and marker style is not optimal in this plot, especially since the points in (b) and (c) are very dense. Is it possible to find a better solution?

- "QC = ..." are very DART-specific statements that only very few readers would understand, please rephrase this in an understandable way.

**Our Response:**

Corrected. (Figure 12 in the revised manuscript, L532)

**Figure 12:**

- The only reference to this figure is in line 406 and that sentence is more or less common knowledge that can be found in various studies. Considering how much information is contained in Figure 12 (4 panels with 3 lines each), the text must contain a deeper analysis of what we can learn from this figure. Otherwise it is not relevant.

**Our Response:**

Thank you for this suggestion. This Figure is designed to illustrate the observation utilization of the WRF/DART-RTTOV system and provide guidelines for observation preparations and model settings.

It is true that some of the conclusions are common knowledge, such as large observation errors correspond to large observation utilization and poor analysis quality (Figure 12, Figure 14). However, Figure 12 provides some unique properties of the WRF/DART-RTTOV system. For example, the outlier threshold is suggested to be 3 for standard DART release. Increasing large outlier threshold value could increase the observation utilization, but it may introduce unstable adjustments to state variables and may destroy the forecast. Therefore, a balance should be maintained between large outlier threshold value and the potentially detrimental effects on forecasts. The analysis and 10 min, 1 hr, and 3 hr first-guess forecasts indicate improved results for larger outlier threshold value. The case study indicates that setting the outlier threshold value to 6 does not cause the collapse of WRF, but generates improvements to the analysis and first-guess forecasts of CWP and cloud coverage. (Figure 13)

Therefore, we made some corrections to the Figure and related sentences in the revised manuscript. (L516-548)

**Spelling, grammar, typos**

Table 1 caption: "data" instead of "dada"
**Our Response:**
Corrected (L298)

L20: There is no need for using semicolons in this sentence. Please replace the semicolons by commas, e.g. "[...] cloud phases, the second [...] positively, and the third..."

**Our Response:**
The Abstract was rewritten, and such errors should be avoided after a major revision.

L20: "The second is the its" -> Chose either "the" or "its"

**Our Response:**
"Its" was replaced by "the assimilation" in the revised manuscript (L24)
L41: It is not common to write it like this. Better would be: "[...] in the study of Vukicevic et al. (2004), model [...]"

**Our Response:**
Corrected. (L67-68)

L45: "[...] while computing [...]"

**Our Response:**
Corrected. (L70)

L194: "are summarized"

**Our Response:**
Corrected. (L289)

L245: "would get the following formula" - that is a strange forumlation.

**Our Response:**
According to the formula (2),

$$\Delta x_{m,n} = \left(\sigma_{p,m}/\sigma_p^2\right)\Delta y_n, \qquad m = 1, \dots, M, n = 1, \dots, N \qquad (2)$$

The ensemble mean of the formula (2) is,

$$\overline{\Delta x_m} = \frac{\sigma_{p,m}}{\sigma_p^2}\left(\overline{y_u} - \bar{y}_p\right) \qquad (2A)$$

Substituting the formula (4), in (2A), we will get formula (5),

$$\overline{y_u} = \frac{\sigma_o^2}{\sigma_o^2 + \sigma_p^2} \bar{y}_p + \frac{\sigma_p^2}{\sigma_o^2 + \sigma_p^2} y_o \qquad (4)$$

$$\overline{\Delta x_m} = \frac{\sigma_p^2}{\sigma_o^2 + \sigma_p^2} (y_o - \bar{y}_p) \qquad (5)$$

Here we denote the observation increment $y_o - \bar{y}_p$ as $R_{inc}$. Therefore, we got the formula (12) in the revised manuscript,

$$\overline{\Delta x_m} = \frac{\sigma_{p,m}}{\sigma_o^2 + \sigma_p^2} R_{inc} \qquad (12)$$

The formulas were reordered in the revise manuscript. The formula (12) in the original manuscript is the formula (11) in the revised manuscript.    (L391)

L250: "is calculated by the following formular"

**Our Response:**
Corrected. (L395-396)

L359: "produced" and not "produce"

**Our Response:**
This part is rewritten in the revised manuscript, and the word "produce" was deleted in the relevant section.

L415: as many observations as possible

**Our Response:**
This part is rewritten in the revised manuscript, and the word "as much as possible" was deleted in the relevant section.

L443: "was detected."

**Our Response:**
This part is rewritten in the revised manuscript, and the word "be detected" was deleted in the relevant section.

**Response to reviewer #2**

**Review of "A preliminary evaluation of WRF (ARW v4.1.1)/DART (Manhattan release v9.8.0)-RTTOV (v12.3) in assimilating satellite visible radiance data for a cyclone case" by Zhou et. al, 2022, submitted**

**General comments**

This paper deals with the data assimilation of visible satellite radiances (also referred to as reflectance). Such observations are relatively new to the numerical weather prediction (NWP) community, since the absence of fast and accurate forward operators made their operational exploitation inconceivable for many decades.

In a set-up using the WRF model, the DART data assimilation framework providing EAKF and RHF filters plus the RTTOV-DOM forward operator, both numerical data assimilation cycle experiments and single observation experiments are conducted in an OSSE framework.

Thereby, relevant aspects related to reflectance data assimilation are shown and outlined, including non-linearity, non-Gaussianity, observation weight related to thinning length scales and update frequencies. Further, limits of reflectance data assimilation are discussed, e.g. the lack of vertical height information of the observed clouds and ambiguities in cloud phase and particle size distribution. While generally interesting, from what I can see, most of these aspects have already been discussed by Scheck et. al, 2020 in the context of the COSMO + KENDA system. I therefore strongly advice that the authors include a detailed discussion of how their findings relate to the previous study, to which extent they confirm or contradict previous findings and which parts of their analysis are uniquely novel.

**Our Response:**

Scheck et al. (2020) assimilated the all-sky satellite visible radiance with an ensemble-based method. Their results lay good foundation for the following research. Our study is similar to Scheck et al. (2020). For example, both of the two studies indicate slightly positive impacts of assimilating visible radiance data on temperature and humidity-related parameters (relative humidity in Scheck et al. (2020) and water vapor mixing ratio in our study). The assimilation is beneficial to the integrals, but cannot improve cloud vertical structures. Nevertheless, there are two general differences between the two studies.

The first is that our study revealed that assimilating the visible radiance data generated slightly positively impacts on horizontal winds, while Scheck et al. (2020) indicated neutral impacts. The second difference is the DA system and model configurations of the two study. Besides the Kalman Filter-based method (EAKF in

this study), the WRF/DART includes a non-Gaussian filters called the Rank Histogram Filter. RHF was tested and compared to the EAKF. The related findings are uniquely novel for this study. In addition, RTTOV is a new capability for DART (quoted from the DART website). The evaluation of the WRF/DART-RTTOV should be meaningful to the further development of the WRF/DART-RTTOV DA system. (L604-620)

While key sensitivities of the data assimilation cycle are discussed and evaluated very detailed with respect to analysis verification, it would be of great practical relevance to understand also how such sensitivities relate to the forecast quality and the forecast error growth. The assumption that a better analysis leads to a better forecast is by no means trivial particularly when dealing with cloud variables whose properties violate the mathematical assumptions of filter algorithms (linearity and Gaussianity), and which are prone to model biases and compensating model errors of the NWP model. Further, the analysis ensemble mean, which is mostly verified in this study, is not a physically consistent state so that it is not obvious to which extent the analysis error reduction related to the ensemble mean is beneficial for the accuracy of the individual forecast ensemble members which are initialised from the respective analysed model states. I therefore suggest to add results related to forecast verification (i.e., forecast verification of experiments 1-6). A discussion of the sensitivities of forecast quality and error growth for cloud variables, but also for other model parameters like temperature and humidity would add significant value to the publication and could provide guidance to colleagues for preparing visible radiance data assimilation also in an operational context.

**Our Response:**
In the original manuscript, the nature run was represented by the ensemble mean of several members to maintain consistency with the ensemble forecasting results for the control run and cycled DA experiments. In the revised manuscript, the nature run was represented by a deterministic forecast. The representation of nature run by a deterministic forecast is commonly used in related fields. In the revised manuscript, the state variables of the nature run are physically consistent. Accordingly, all the cycled DA experiments were re-performed by assimilating synthetic visible radiance data simulated based on the model state of the nature run. (L152-168)

It is true that better analysis des not ensure better forecast. In the original manuscript, we did not present first-guess forecasting results. In the revised manuscript, both the analysis and first-guess forecasts of cloud variables (cloud fraction and cloud water path) and non-cloud variables (water vapor mixing ratio, perturbation potential temperature, and the x- and y-wind components) were analyzed. Sensitivities of the first-guess forecast quality and error growth for the cloud variables and non-cloud variables to different model settings and observation preparations were discussed, including the sensitivities to filter algorithms, cycling variables, outlier threshold values, observation errors, and observations with or without thinning. (L453-548)

This is done by comparing different experiment groups containing different cycled DA experiments. There altogether fourteen cycled DA experiments in the revised manuscript. (L298-304)

While this manuscript contains some very interesting material, to be suitable for publication it requires some substantial changes. Please find below.

- The key research questions should be stated more clearly in the paper overview and at the beginning of each paragraph. It should be motivated why the different investigations are done and which question is followed therein.

**Our Response:**

Since RTTOV is a new capability for WRF-DART (quoted from the DART website *https://docs.dart.ucar.edu/en/latest/guide/Radiance_support.html?highlight=RTTOV*), we hope to evaluate the performance of the WRF/DART-RTTOV system under different model settings to answer the following questions. 1) What is the advantages and limitations of assimilating the FY-4A VIS imagery to the forecast of a tropical storm case by using WRF/DART-RTTOV system? 2) What is the better choice for the WRF-DART/RTTOV model settings and observation preparations? 3) What are the future works for the real DA applications of FY-4A satellite visible radiance data? These research questions were stated in Section 1 of the revised manuscript. (L119-125).

In this study, single observation experiment and cycled DA experiments were performed to explore different aspects of assimilating the visible radiance data. The single observation experiments is designed to demonstrate the basic technical functionality of assimilating visible radiance data, and to reveal the potentials, inabilities and ambiguities of assimilating VIS radiance data. (L266-269). The cycled DA experiments were performed to evaluate the influences of different model settings and observation preparations on the analysis and the first-guess forecast when VIS radiance data was assimilated. The purpose of these cycle DA experiments is to reveal the forecast quality and error growth of the forecasting errors for main state variables. We think the related results could provide some guidance to the settings of WRF/DART-RTTOV and an outlook to the future works (L283-285). The cycled DA experiments include fourteen experiments with different set ups. The purpose of each experiment group was illustrated in section 2.3.3 (L290-297).

In the Section 3, the main purpose of each paragraph was added to the revised manuscript to make the results clear (L454-456, L473-475, L482-487).

- Could you motivate why you do have chosen to assimilate the observations in physical variables of radiance rather than reflectance which makes it much more easy to estimate how optically thick the clouds are? And which seems to be much more convenient in the data assimilation community?

**Our Response:**

DART provided many observation types for different observations. For the assimilation of FY-4A/AGRI data, the supported observation type only includes "FY4_1_AGRI_RADIANCE", which denotes the radiance data for FY-4A/AGRI. For this convenience, we choose to assimilate the visible radiance data rather than the bidirectional reflectance (BRF), despite the latter is more convenient to estimate how optically thick the clouds are.

We think assimilating the two quantities are equivalent because radiance could be converted to reflectance by the formula,

$$r(-\mu, \mu_0, \varphi - \varphi_0) = \frac{\pi L(-\mu, \mu_0, \varphi, -\varphi_0) \cdot d^2}{\mu_0 E_{0,b}^{std}}$$

where r denotes the BRF as a function of viewing and solar angles, L denotes the radiance, μ denotes the cosine of viewing zenith angle, $\mu_0$ denotes the cosine of solar zenith angle, $\varphi - \varphi_0$ denotes the different between the solar and viewing zenith angles, $E_{0,b}^{std}$ denotes the standardized band-weighted solar irradiance, and d denotes the solar distance in AU.

We think that the reflectance data seems more convenient for real DA applications between it is provided by may satellite centers more commonly than radiance data. Therefore, adding the reflectance to the supported observation type list should be a future work of WRF-DART/RTTOV system.

- The fundamentally new methods and findings for the research community should be pointed out in a more precise way

**Our Response:**

We split the abstract into two paragraphs. The first paragraph described the WRF/DART-RTTOV models, general purposes and methods of this study. The second paragraph was organized following the order: advantages by the assimilating the VIS radiances; suggestions to the model settings and observation preparations; limitations of the assimilation; and future works. We think that the revised manuscript should present the new methods and findings more precisely. (L8-28)

- Forecasts should be added to the cycled data assimilation experiments to show the sensitivities of both analysis and forecast error to the update frequencies, thinning length scales, chosen DA filter and so forth

**Our Response:**
Corrected. (Figure 9; Figure 10; Figure 11; Figure 13; and Figure 14).

- The figures have to be revised fundamentally. Please increase the picture resolution and enlarge lines, labels and axes. It is very hard to differentiate

between the triangles, diamonds and squares even when viewing the figures with a large zoom factor.

**Our Response:**
Corrected throughout the manuscript. The figures throughout were revised with a resolution of 300 dpi.

- The questions below should be answered in the text

**Our Response:**
Corrected.

- Please consider the comments below to improve the text

**Our Response:**
Corrected.

**Specific comments**

- Title: Code versions do not have to be part of the title of a scientific paper, suggestion: "A preliminary evaluation of visible radiance data assimilation for a cyclone case"

**Our Response:**
Corrected. In the revised manuscript, the title was modified as "A preliminary evaluation of visible radiance data assimilation by the WRF/DART-RTTOV system for a tropical storm case". (L1-2)

**Sec 1.**

- Please add some comments on challenges and potential of all-sky data assimilation

**Our Response:**
Corrected. (L32-47)
All-sky MW data has been operationally assimilated at some Numerical Weather Prediction (NWP) centres (Bauer et al., 2010; Zhu et al., 2016). However, operational DA of all-sky MW data is limited to humidity- and temperature-sounding channels (Carminati and Migliorini, 2021) because MW radiance at these channels is insensitive to surface emissivity and skin temperature, whose accurate estimates are challenging under cloudy skies (Hu et al., 2021). DA of MW data is also challenging to separate the radiance contribution form cloud and non-cloud variables (especially temperature and humidity) (Geer et al., 2017). The assimilation of all-sky IR radiance

data also shows beneficial aspects. Existing studies suggest positive effects on water vapour and temperature by assimilating the IR data in clear sky (McCarty et al., 2009; Ma et al., 2017). In addition, DA of IR radiance data in cloudy regions shows improved analysis of column integrated water and improved forecasting skills in the mid- and upper-troposphere (Stengel et al., 2013; Geer et al., 2019). However, the assimilation of IR radiance data in cloudy regions is still complicated by the non-linear relationship between the observation and state variables and the related non-Gaussian problems (Li et al., 2022), the difficulty to separate cloud signals and non-cloud signals (Geer et al., 2017), and the difficulty to constrain the layered structures in multi-layer clouds (Prates et al., 2014).

- Please elaborate bit more on why you think it is interesting to assimilate visible satellite radiances. Which forecast impact do you expect? What is different from IR or MW all-sky data assimilation?

**Our Response:**
The reason why we think assimilating the visible radiance data is interesting include the following 3 aspects. 1) Visible radiation can penetrate a certain depth of cloud fields. In comparison, satellite IR data only provide information on cloud top microphysics. 2) As a complement to precipitation radar that is much more sensitive to large hydrometeors or precipitation particles, which usually occurring at the mature and developing stages of convective systems, VIS radiance data are closely related to small cloud droplets. 3) VIS data usually have higher spatial resolution than MW data. Therefore, high-resolution satellite VIS radiance data provide information on cloud properties with a great significance for cloud-resolving model simulation. (L51-58)

The forecast impacts we are expecting is to improve the forecast quality of cloud variables, including cloud water path, cloud effective radius, cloud fraction, and precipitation. (L62-63)

The difference of all-sky DA of VIS data including the two following two aspects. 1) Compared with the MW and IR data, the radiance contribution from cloud could be easily extracted from the VIS observations because the VIS radiance data is much more sensitive to cloud variables than non-cloud variables. 2) VIS radiance is not sensitive to cloud layout in the vertical directions, but to the accumulated cloud properties. This makes the assimilation much easier for heterogeneous cloud scenarios. (L58-62)

- Please note and correct in the text: RTTOV is no forward operator, but rather a collection of forward operators (radiative transfer package)

**Our Response:**
Corrected throughout the revised manuscript.

- The goal of the publication and its value to the scientific community has to be stated clearly at the end of the introduction section

**Our Response:**

The goal of this study is to evaluate the performance of the WRF/DART-RTTOV system under different model settings to reveal the advantages and limitations of assimilating the FY-4A VIS imagery to the forecast of a tropical storm case, and to provide basic guidelines or suggestions to the model settings and observation preparations for the future DA applications of FY-4A VIS radiance data in real cases. The results should be representative to the upcoming FY-4B VIS radiance data because the designs of the AGRI payload for the two satellites are similar. (L119-125)

**Sec 2.1**

- Are the grid spacings between nature run, control run and DA runs equivalent?

**Our Response:**

Yes it is. WRF model domain settings are the same for the nature run, the control run, and the cycled DA experiments to avoid errors due to displacement of grids between the observation and simulations. (L141-142)

- Why do you use higher resolved LBCs for the control / Da experiment than for the nature run / truth? An OSSE should represent the difference between real atmosphere (which is much higher resolved than a forecast model) and a forecast model. Thus, I would rather use the higher resolved LBCs for nature run / truth. Do you conduct short-range forecasts as well? Do the LBCs introduce the cyclone to the model domain across the lateral boundaries or does it fully develop within the model domain? If the first is the case this may be problematic.

**Our Response:**

To avoid discontinuities and poor results at the boundary, LBCs at each analysis time were updated based on the analysis and WRF lateral boundary conditions by an approach built in the DART *pert_wrf_bc* module. This is the reason why we choose the higher resolved LBCs, as we will do in real DA applications, for the DA experiments than for the nature run. (L173-176)

We did short-range forecast for the nature run, the control run, and cycled DA experiments. The nature run was initialized by a cold start at 12:00 UTC 17 August, 2020. After a spin-up time of 14 hr, the state variables provided a proxy true atmosphere (L161-163). The control run and the cycled DA experiments were initialized at 00:00 UTC 18 August, 2020. After a spin-up time of 2 hr, synthetic visible radiance observations were assimilated to the ensemble members. In each DA

cycle, a short-range forecast for each of the ensemble members was conducted, with the forecasting time span of 10 min, 1hr, and 3 hr, depending on different experiment designs. (L177-180)

The nature run is designed to describe the bulk properties of a tropical storm named Higos, which is originated from a tropical disturbance occurred over the north of Luzon, the Philippines on August 16, 2020. The system tracked northwest toward the South China Sea, and intensified into a tropical storm on 19:00 August 17. The tropical storm was further developed into a typhoon system on 12:00 UTC, August 18. The model settings for the nature run did not introduce the system cross the domain boundaries, but within the domain near the northwest of Luzon Island. The nature run captures the bulk track properties of the tropical storm. Therefore, we thick the nature run has certain representativeness to the real system.(L132-137)

- Why do you use the ensemble mean of the nature run as truth? The resulting model state is physically inconsistent between the variables

**Our Response:**

In the original manuscript, the nature run was represented by the ensemble mean of several members to maintain consistency with the ensemble forecasting results for the control run and cycled DA experiments.

In the revised manuscript, the nature run was represented by a deterministic forecast. The representation of nature run by a deterministic forecast is commonly used in related fields. After revision, the state variables of the nature run are physically consistent. Accordingly, all the cycled DA experiments were re-performed by assimilating synthetic visible radiance data simulated based on the model state of the nature run. (L152-168)

- How do you exactly produce the synthetic observations? What is the role of the 2km- original AGRI observations? Do you use them as observation locations for the synthetic observations? How do you assign observations and model grid points? Do you first assign model grid points to observation locations (which will lead to one grid point being assigned to many observations at a 15 km – 2km scale difference) and then apply thinning at the observation locations? Do you interpolate and how or do you use nearest-neighbour? Is it right that you simulate the observations based on the truth which is equal to ensemble mean? The next reasonable step would be to perturb the synthetic observations based on an estimated observation error distribution. Do you do that and how?

**Our Response:**

The synthetic visible radiance observations were generated by the RTTOV radiatve transfer package for the FY-4A/AGRI channel 2 settings based on the state variables provided by the nature run. We realized that assimilating the observations

derived from the ensemble mean is problematic in the original manuscript. Therefore, the theoretical truth, represented by the state variables of the nature run, is simulated by a deterministic forecast in the revised manuscript (L152-168). The methods to simulate synthetic visible radiances are detailed in 2.2. (L186-211)

This study is performed in an OSSE framework, no 2km-original AGRI observations were used. Instead, it is the simulated 15km-resolution radiance data that were assimilated. WRF model domain settings are the same for the nature run, the control run, and the cycled DA experiments with a grid spacing of 15 km. The simulated VIS imagery was approximately equivalent to the superobbing of the 2km-resolution imagery, as is provided by the FY-4A observations, by averaging the 2 km × 2 km imagery for every block of about 7×7 pixels. Because the observation locations and model grid points are overlapped, the locations of the synthetic observations are directly assigned to the model grid points without interpolation during the assimilation processes. (L141-142, L164-168)

The study involves the EAKF and RHF methods. An observation error $\sigma_o$ was assigned to a radiance observation $r_o$ before the assimilation. Both EAKF and RHF assume that the observational error variance is Gaussian so that the observational likelihood is Normal $(r_o, \sigma_o^2)$. Unlike EnKF, which is a perturbed observation filter and that perturbations were added to the observations before assimilation, the EAKF and RHF are two variants of deterministic filters which do not add perturbations to the observations. (Section 2.3.1, L232-262)

- Please explain the set-up of the OSSE in much more detail following the questions posed above. At the current state, it is hard to understand and not reproducible.

  **Our Response:**
  Section 2 is reorganized in the revised manuscript according to the comments posed above, many details were added to the set-ups of the OSSE. (Section 2.1, L141-184; Section 2.3.2 and 2.3.3, L264-304)

- Please explain in the text if you assimilate any other observations or only visible radiances

  **Our Response:**
  We only assimilated the simulated visible radiances at a resolution of 15 km, and this information is declared at the beginning of section 2. (L130-131)

**Sec 2.2**

- For better understanding of the sensitivities of visible radiances on model variables please add a discussion of the subgrid-scale cloud variables which

are presumably input to the forward operator. How is that realized in your system? How are subgrid-scale clouds parameterized? This may be important to discuss and understand the potential detrimental impact on non-cloud prognostic variables

**Our Response:**
The inputs of RTTOV include cloud fraction (CFC). For the WRF model configurations in this study, CFC is a subgrid-scale cloud variable. The CFC parameterization depends on the grid-scale relative humidity (RH), the saturation water vapour mixing ratio ($q^*$), and cloud water + ice mixing ratios ($q_{l+i}$) (Xu and Randall, 1996),

$$\text{CFC} = \begin{cases} RH^p \left[1 - \exp\left(\frac{-\alpha q_{l+i}}{[(1-RH)q^*]^\gamma}\right)\right], if\ RH < 1 \\ 1, if\ RH \geq 1 \end{cases}$$

where $p$, $\alpha$, and $\gamma$ are suggested to be 0.25, 100, and 0.49 separately.

Therefore, an implicit relationship between VIS radiance and RH should be expected due to the parameterization of CFC. Given that RH not only depends on the water vapor mixing ratio (Q), but also on temperature (T) and pressure, spurious covariance between VIS radiance and Q/T was generated due to the ensemble spread of Q/T (different Q and T for the ensemble members would blur the relationship between VIS radiance and Q/T). As a result, neutral impacts on Q and T were revealed for the single observation experiments. Because RH is not a state variable for WRF model, the results on RH were not presented explicitly. (L192-196; L343-355)

Nevertheless, positive impacts on T, Q, and horizontal winds were revealed for the first-guess forecast and analysis state variables. The beneficial impacts on non-cloud variables were generated due to positive feedbacks to cloud variables by condensation, evaporation, freezing, and cloud-radiation interactions. (L482-515)

- Please define "cloud water path" in the text (appears for the first time in Figure 2)

**Our Response:**
CWP denotes the vertically integrated cloud liquid and ice water mixing ratio in an atmospheric column, which is calculated by the following formula,

$$\text{CWP} = \int_{P_s}^{P_t} \frac{1}{g}(Q_c + Q_i)dP$$

where $P_s$ and $P_t$ denote surface and at the model top pressure, Qc and Qi the liquid water mixing ratio (the sum of the mixing ratio of cloud droplet and rain) and ice water mixing ratio (the sum of the mixing ratio of ice, snow, and graupel), and g the gravitational acceleration (9.8 ms$^{-2}$). (L213-218)

**Sec 2.3.2**

- It will help the reader if you explain in more detail the goal of the pointwise DA experiments. What do you want to show here?

**Our Response:**
The single observation experiments assimilate an observation at a targeting pixel, and the adjustments of state variables at the targeting pixel are only caused by the assimilation of one observation. Therefore, it is convenient to demonstrate the basic technical functionality of assimilating visible radiance data, with the purpose to reveal the potentials, inabilities and ambiguities of the assimilation. The focusing cloud variables include Qc, Qi, Re, and CFC. The focusing non-cloud variables include water vapour mixing ratio (QVAPOR), perturbation potential temperature (T), and the x- and y-wind components (U and V). (L266-271)

- What is the meteorological situation at the 4 points that you have chosen in the domain? Please motivate why you have chosen exactly these points

**Our Response:**
We made many revisions to the original manuscript. In the revised manuscript, three special points were selected for further analysis.
Point 1 refers to an ideal case where the posterior is within the range bounded the the prior estimate and the truth both in the cloud variable space and in the observation space. Point 1 is selected to demonstrate the beneficial impacts of the assimilation on CWP, and the vertically averaged CFC and Re. In addition, the inabilities to improve cloud vertical structures and cloud phases were illustrated. The meteorological situation of Point 1 was added in the revised manuscript. (L358-384)
Point 2 refers to a case where the posterior is within the prior estimate and the truth in observation space but not in cloud variable space. The negative impacts on CWP are mainly caused by the non-linear relationship between CWP and VIS radiance. Therefore, point 2 is selected to demonstrate the limit of the assimilation due to non-linear problem. (L385-402)
Point 3 refers to a case where the posterior is beyond the prior estimate and the truth both in observation space and in cloud variable space. The negative impacts on observation are mainly caused by the non-Gaussian properties of the prior distribution. Therefore, point 3 is selected to demonstrate the limit of the assimilation due to non-Gaussian problem. (L403-411)

- Would it be an idea to refer to the pointwise data assimilation experiments as single observation experiments? This is a term that seems to be more convenient in the data assimilation community

**Our Response:**

Corrected throughout the manuscript.

- Please clarify that the cycled experiments are also run in OSSE set-up. This does not seem to become clear in a moment.

**Our Response:**
Corrected. (L421)

- Do you also run forecasts or only DA experiments?

**Our Response:**
We only run DA experiments for single observation experiments (L269). Forecasts were performed for the cycled DA experiments (L281-286).

- VIS data assimilation strongly interacts with forecasts, so I would ask you to verify forecasts as well to show if the DA is successful in terms of forecast impact

**Our Response:**
It is true that VIS data assimilation strongly interacts with forecasts. The impacts on the first-guess forecast were discussed for the cycled DA experiments in Section 3.2. The cycled DA experiments include many experiments with different OSSE set-ups, which we think could show the positive or negative impacts on cloud and non-cloud variables.

- Table1: Please explain the variable names or rather write the physical variable names, e.g. QICE=cloud ice mixing ratio.

**Our Response:**
Corrected. (L298-304)

**Sec 3.1.**

- I would suggest to refer to that kind of experiment as "single observation experiment"

**Our Response:**
Corrected throughout the manuscript.

- Please explain what you mean by "cloud water path". Is that only vertically integrated liquid water? Or liquid or ice water? Or the sum of the two? May I ask why you do not show the ensemble distribution of cloud water and cloud ice in the left-hand plots since you compare to them on the right-hand side?

**Our Response:**

CWP denotes the vertically integrated cloud liquid and ice water mixing ratio in an atmospheric column, which is calculated by the formula,

$$\text{CWP} = \int_{P_s}^{P_t} \frac{1}{g}(Q_c + Q_i)dP$$

where $P_s$ and $P_t$ denote surface and at the model top pressure, Qc and Qi the liquid water mixing ratio (the sum of the mixing ratio of cloud droplet and rain) and ice water mixing ratio (the sum of the mixing ratio of ice, snow, and graupel), and g the gravitational acceleration (9.8 ms-2). (L213-218)

The VIS radiance data depends on the sum of the cloud water and cloud ice (CWP). Therefore, the assimilation could adjust CWP correctly, but could not improve the ice water or cloud water separately. Therefore, the left-hand shows what the assimilation can improve, and the right-hand shows what the assimilation cannot improve.

To avoid misleading, we replaced this figure by Figures 3 and 4 in the revised manuscript. We hope that the two figures could illustrate the beneficial impacts (on CWP, Re, CFC) and the inabilities (to improve cloud vertical structures and cloud phases) in a better way. (L350, L367)

- Please motivate more for the reader why you assess and show that kind of experiment. What is your goal with that? Please make that very clear to explain the key issue with ambiguities in visible radiances, i.e. total water mass, cloud phase, effective radii, vertical position, multiple layers with different phases

**Our Response:**

We show the single observation experiments for three main purposes. (L334-339)

The first purpose is to demonstrate the beneficial impacts of the assimilation both in the observation and state variable space. The results indicated beneficial impacts of the assimilation on the total water mass in one column (CWP), the effective radius profile across one column, and the cloud fraction across one column. In addition, the results indicated the ambiguities to improve cloud vertical structures and cloud phases for a multi-layer and mixed-phase cloud case. This is illustrated by Point 1. (L356-384)

The second is to demonstrate beneficial impacts in observation space but neutral or negative impacts in state variable space. This is mainly caused by the non-linear problem, which is illustrated by Point 2. (L385-402)

The third purpose is to demonstrate the negative impacts of the assimilation due to non-Gaussian problem. This is illustrated by Point 3. (L403-411)

- In Figure 3, do you work in observation space on the left-hand side and in model space on the right-hand side? So you try to figure out to what degree the model variables are improved if the analysis is drawn towards the observation in observation space? Could you clarify that in the text, please?

**Our Response:**
Exactly. Changes in observation space could introduce changes in state variable space. Figure 3 in the original manuscript only presents results in observation space and in cloud variable (CWP and liquid/ice water mixing ratio) space. The results for Re, CFC, and non-cloud variables were not presented. Therefore, this part was totally changed in the revised manuscript.

In the revised manuscript, the comparison between first guess and analysis of cloud and non-cloud variables are presented by Figure3 and Figure 4. We hope that changes in observation space and model variables' space were better represented by Figures 3 and 4 in the revised manuscript. (L350, L367)

- I could not distinguish between the lines in Figure 3. Please draw fatter lines, fatter axes, fatter labels. Use different colours rather than symbols because it is unfortunately really hard to distinguish between them. It is confusing that the diamond sign means "truth" on the left and "analysis" on the right

**Our Response:**
Corrected here and elsewhere.

- Since GMD is a journal dealing with models and the key goal of data assimilation is better meteorological forecasting skill it will add great value to the paper if you discuss not only the statistical properties of the single-obs experiments, but also explain the meteorological situations: e.g. in Figure b1/b2) in the truth run we have an optically thick water cloud. However, the model shows an ice cloud lying over a water cloud. Data assimilation draws radiance towards the truth and is well able to enhance the water cloud. However, the false alarm ice cloud is also enhanced, since a) VIS observations can only constrain vertical integrals, b) there seems to exist spurious correlation of the ice clouds in the background ensemble with the observation and we cannot vertically localize that due to missing information on the cloud top height and vertical cloud extent. Clarify that VIS observations are sensitive to the cloud water mass in the column and the particle size distribution. Ice clouds consist of few big particles and are typically much more optically thin

than water clouds that consist of many small particles. Try to explain a bit more the microphysical connection between clouds and radiance to the reader.

**Our Response:**

Details on the meteorological situations for Point 1 were added in the revised manuscript. The point 1 corresponds to a one-layer ice cloud between 400 ~ 200 hPa with a CWP of 0.01 kg m-2 and a top-of-atmosphere (TOA) radiance of 3.63 mW m-2 Sr-1 for the nature run. The first-guess forecast simulated a two-layer mixed-phase cloud with a false alarm liquid water cloud simulated below 500 hPa. The first-guess ensemble mean CWP and equivalent VIS radiance is 1.33 kg m$^{-2}$ and 7.29 mW m$^{-2}$ Sr$^{-1}$, respectively. After assimilating the satellite VIS radiance data, the first guess was drawn toward the nature run both in the observation space, with a decreased ensemble mean radiance of 6.40 mW m$^{-2}$ Sr$^{-1}$, and in the CWP space, with a decreased ensemble mean CWP of 0.85 kg m$^{-2}$. As a result, Qc, Qi, CFC, and Re were adjusted collaboratively toward the nature run. (L358-364)

The synthetic VIS radiance observation is not sensitive to cloud vertical structures but to the accumulated cloud water/ice mass. As a result, it is difficult to correct cloud vertical location errors due to a lack of information on cloud top height and vertical cloud extent. In addition, the assimilation cannot remove the false alarm liquid clouds due to the spurious covariance between the VIS radiances and liquid water clouds in the background. (L372-375)

The analysis increment of each state variable is linearly related to its covariance with observation. Therefore, the vertical structures and phases of the posterior estimate are mainly determined by those of the prior estimate. Larger first-guess estimate of the state variable would generate larger covariance, and larger adjustments to the first guess should be expected. Because VIS radiance is positively related to Qc/Qi, the adjustments of Qc/Qi were much more distinct for the lower layer than the upper layer ($\leq$ 300 hPa) due to larger Qc and Qi in the lower layer in the background ($\geq$ 400 hPa). Similar results were also found for Re, with larger liquid water particles occurred in the middle layer (~ 600 hPa) and smaller liquid water particles occurred in the lower layer (~ 800 hPa). (L375-384)

- Please add this kind of meteorological detail to the clarify the potential and limits of the VIS data assimilation in view of specific meteorological cloud situations

**Our Response:**

Details on the meteorological situations for Point 1 were added in the revised manuscript. (L358-364)

Assimilating the visible radiances has the potential to improve the forecasting of accumulated cloud water/ice mass, and the vertically mean cloud variables. However, neither can the assimilation correct cloud vertical structures, nor can the assimilation improve cloud phases, especially for multi-layer and mixed-phase clouds. (L356-366, L372-375)

- In Scheck et. al 2020 such kind of case study has been performed as well. Please reference and compare your discussion to the results found there.

**Our Response:**

The positive and negative impacts, the inabilities to correct cloud vertical location errors, and the ambiguities to improve cloud phase analysis by assimilating the VIS radiance data agrees well with Scheck et al. (2020).

There are some differences as for the impacts on non-cloud variables (temperature, water vapor mixing ratio, and horizontal winds). Our study indicates that the assimilation generates positive impacts on temperature and water vapor mixing ratio for the first-guess forecast and analysis, the results conformed with Scheck et al. (2020). However, slight positive impacts on horizontal wind speeds were demonstrated in this study but Scheck et al. (2020) reported neutral impacts. We ascribe the slightly positive impacts on horizontal winds to the feedbacks to the convergence or divergence related to the "radiance——cloud——vertical velocity——convergence and divergence——horizontal winds" relationship. This relationship should differ in weather systems. Therefore, the different impacts on horizontal wind speeds revealed by the two studies could be caused by different characteristics of the weather systems. (L604-615)

**Sec 3.1**

- Do you assimilate any observations additional to radiance observations here?

**Our Response:**

Only the VIS radiance observations were assimilated throughout the paper. (L130-131)

- Do you assess first guesses or analyses or forecasts here?

**Our Response:**

For the single observation experiments, only the analyses and first guesses were analyzed. (L269)

The first-guess forecasts were analyzed for the cycled DA experiments (Section 3.2, L281-286)

- Please improve Figure 7. It is very hard to distinguish the symbols. Use fatter labels and axes and fatter lines and maybe different colors for the different experiments

**Our Response:**
Corrected here and elsewhere.

- Clarify if the cycled experiments are OSSE experiments or if you assimilate original satellite observations

**Our Response:**
This study is performed in an OSSE framework. Namely, we did not assimilate the original satellite observations but the simulated synthetic visible radiances. (L130-131)

- You should add that the no-clouds situation in the ensemble is referred to as "zero-spread" problem

**Our Response:**
Corrected. (L383, L449, L567, L601, L652)

- You state that RMSE, MAE etc. measure different aspects of accuracy than FSS. Please explain which aspects and why you want to assess them

**Our Response:**
The Root Mean Square Error (RMSE) and Mean Absolute Error (MAE) are two of the most commonly used metrics to assess the simulation errors (Kurzrock et al., 2019). Compared with MAE, RMSE is much more sensitive to extremely large errors. For satellite VIS radiance assimilation, extremely large analysis increments of CWP were rarely expected (details provided in Section 3), implying that the differences of RMSE for the first-guess and for the analysis model state was not as distinct as MAE. In order to make more clear the influences of assimilating the VIS radiance data, MAE is used to measure the difference between the simulated CWP and the theoretical true CWP (derived from the nature run). (L306-315)
The fraction skill score (FSS) is used to measure cloud location errors in the horizontal directions. (L316-320)

- To which degree do the results found for the six different experiments hold for forecast impact?

**Our Response:**
The revised manuscript included 14 cycled DA experiments. In general, beneficial impacts were revealed for CWP, cloud coverage, non-cloud variables, and precipitation. The first-guess forecasts results were presented by Figure 7, 9, 10, 11, 13, 14, and 15.

- Please replace "fake" correlations by "spurious" correlations

**Our Response:**

Corrected.

- Please elaborate a bit on why less observations and less frequent update intervals may lead to better overall forecasting skill

**Our Response:**

We made a terrible mistake in the original manuscript that the first guess for the next cycling step was not updated correctly. This mistake was fixed in the original manuscript, and positive impacts on cloud variables and slightly positive impacts on non-cloud variables were revealed. Therefore, better overall forecasting skill corresponds to more observations (larger outlier threshold value, observations without thinning) and small update intervals (10 min update is better than 1 hr and 3 hr). (Section 3.2 in the revised manuscript)

- Do you have suggestions why VIS DA may have detrimental impact on dynamic and thermodynamic prognostic variables? What is the role of the NWP model in here? What is the role of subgrid-scale clouds? Could you add a plot illustrating the detrimental impact please? Please discuss this issue in a bit more depth and to debate potential fixes

**Our Response:**

As indicated above, we made a terrible mistake in the original manuscript that the first guess for the next cycling step was not updated correctly. Therefore, odd results were revealed in the original manuscript. The problem is fixed in the revised manuscript, and slightly positive impacts were introduced to the dynamic and thermal state variables.

The role of NWP model is to adjust the cloud and non-cloud variables by evaporation, condensation, freezing, and cloud-radiation interactions with model integration. In general, adjustments to cloud variables by the WRF model counteracts with the assimilation, and the adjustments to non-cloud variables by the WRF model improve the forecast quality of non-cloud variables. (L491-515)

The VIS radiance is closely related to the subgrid-scale clouds, which is parameterized by cloud fraction (CFC). Assimilating the VIS radiance data improved the analysis quality of CFC. The improved CFC will introduce positive impacts on non-cloud variables. For example, decreased CFC for the analysis state will enhance the direct radiation flux on the surface layer, increasing the low-level temperature toward the truth. Therefore, subgrid-scale clouds will introduce positive impacts on the analysis and forecast quality.

- I am unhappy with your term "thermodynamic" variables. Wind is no thermodynamic variable. Maybe you could refer to the variables you want to address as "non-cloud" variables

**Our Response:**
Corrected throughout the manuscript.

- You state that VIS radiance data do not have an apparent dependence on "thermodynamic" variables. I do not agree with that. Clouds are advected by wind fields so that cloud position error is correlated with wind field errors. Clouds depend on temperature and humidity. Subgrid clouds are typically parameterized in terms of grid-scale humidity fields.

**Our Response:**
Thank you for pointing this out. We agree with you that VIS radiance data should be related to atmosphere thermal and dynamical variables. Therefore, the sentence was corrected in the revised manuscript "the VIS radiance is insensitive to U and V at the analysis time". (L341-342)
The dependence of the VIS radiances on cloud fraction, and on the water vapor mixing ratio and temperature were also discussed with more details in the revised manuscript. (L340-349, L491-515)

**Sec 3.2.1 (I think this part should refer to 3.2.2 in the original manuscript)**

- At first glance it has been unclear to me which kind of model state you verify here. Is it linear analyses, nonlinear analyses, forecasts, first guesses? Please clarify both in the text and in the figures

**Our Response:**
We were verifying the effective radius of liquid water clouds (Re) in this section. Figure 9(a) gives the results for the MPE of Re for the analysis field, and Figure 9(b) gives the results for the MPI of Re for the analysis increments (posterior - prior). We wanted to demonstrate the impacts of assimilating VIS radiances on Re. In the revised manuscript, the impacts on Re were shown by Figure 3 and Figure 4 to avoid confusions. (L350, L367)

- Please explain why it is interesting to assess the temporal evolution of MPI / MPE of effective radius

**Our Response:**
Re is a 3D state variable. It is more convenient to assess the assimilating impacts on Re by 2D variables such as MPI and MPE. What we wanted to present is that small

impacts on Re were generated (Figure 9(a)), and that large forecast errors of Re cannot be effectively reduced by assimilating the VIS radiances. In the revised manuscript, the impacts on Re were shown by Figure 3 and Figure 4 to avoid confusions. (L350, L367)

- Please use the term "false alarm" clouds instead of "fake" clouds

**Our Response:**
Corrected throughout the manuscript.

- Please replace "updated in negative ways" by "analysis increments with wrong signs / wrong magnitudes"

**Our response:**
We changed the word to "positive/beneficial impacts" or "negative impacts" in the revised manuscript.

- Does the underestimation of the effective radius come along with an overestimation of radiance, i.e. a positive radiance bias?

**Our Response:**
We analyzed this topic in section 3.1 in the revised manuscript. Assimilating the VIS radiance data generated beneficial impacts on Re and radiance in most cases. The VIS radiance data is not only negatively related to radiance, but also positively related to cloud fraction (CFC) and the accumulated cloud water/ice mass (CWP). Therefore, it is not necessarily true that the underestimation of Re come along with an overestimation of radiance. For example, more cloud water/ice corresponds to larger cloud hydrometeors (Figure 4). Therefore, larger Re corresponds to larger radiance, and a positive covariance between Re and VIS radiance is generated. As a result, a negative Re bias comes along with a negative radiance bias. (L358-366)

- Is the effective radius input to the RTTOV-DOM forward operator? Could you motivate why you show Figures 9 and 10, please? What is shown in Figure 9?

**Our Response:**
Yes it is. The effective radius of liquid water clouds (Re) is an input for RTTOV-DOM forward operator. Since the selected ice optical property scheme for RTTOV has no explicit dependent on the effective radius of ice particles, the effective radius of ice clouds is not an input to RTTOV-DOM. (L199-204)
Re is a 3D state variable. It is more convenient to assess the assimilating impacts on Re by 2D variables MPI and MPE. What we wanted to present is that small

impacts on Re were generated (Figure 9(a)), and that large forecast errors of Re cannot be effectively reduced by assimilating the VIS radiances. Figure 10 gives an example to test the robustness of the results in Figure 9.

In the revised manuscript, the impacts on Re were shown by Figure 3 and Figure 4 to avoid confusions. (L350, L367)

- Since you seem to assimilate the VIS observations over high terrains in China – do you have any quality control included that rejects observations that may be mixed up with snow or ice?

**Our Response:**

The whole study is performed in an OSSE framework. The surface type is already known before assimilation, and was directly assigned to the locations where observations were assimilated. Therefore, we did not include quality control method to identify observations mixed up with snow or ice for current study.

Some of the observations were rejected due to non-monotonic pressure, i.e., pressure increases with altitude, for the prior ensemble members. The non-monotonic pressure was generated at some points during the interpolation of the perturbed first-guess model state. The results indicate that the performance of the WRF-DART system is slightly prohibited over complex terrain regions.

- How do you set the observation error? Next to the number of observations assimilated which is determined by thinning length scales and update frequencies you can control the weight of the observations by choosing a larger observation error. It would be nice if you included that in the discussion and potentially even in your experiments

**Our Response:**

Three observation errors of 1, 2, and 4 were assigned to different experiments. Three update frequencies of 10 min, 1 hr, and 3 hr were assigned to different experiments. In addition, observations with a thinning length scale of 60 km were assimilated to a cycled DA experiment. The impacts of different model settings and observation preparations on the analysis and first-guess forecast were analyzed in the revised manuscript. (Table 1, L304)

**Sec 4.**

- What is the main message that you want to present to the reader related to your discussion on observation rejection?

**Our Response:**

Because this study is an evaluation of the WRF/DART system, we want to explore the utilization of the observations. A purpose is to find the reasons why some of the observations were rejected and to correct the rejections. The results indicate that some of the observations were rejected by the DA system due to the non-monotonic pressure over complex terrain areas such as the Qinghai-Tibet Plateau, Tianshan Mountain, and Central Taiwan ranges. The results indicate that the performance of the WRF/DART-RTTOV system is slightly prohibited over complex terrain regions. Other observations were rejected due to the differences between the observations and the prior ensemble mean equivalent observations are too large. Increasing the outlier threshold value to 6 increased the observation utilization and does not cause the collapse of WRF, but generates improvements to the analysis and first-guess forecasts of CWP and cloud coverage. Therefore, a larger outlier threshold value larger than 3, which is the suggested outlier threshold value by the standard DART release, is suggested for the assimilation of VIS radiance data (L516-535)

- Please revise Figure 11. It is very hard to distinguish between the triangles, squares etc. and to recognize them. Maybe you do not have to display every time step in the plot

  **Our response:**
  Corrected (Figure 12 in the revised manuscript)

- I'm wondering why the departure between first guess and observations increases over time. In a healthy DA system, the average first guess error tends to decrease with increasing number of DA cycles. Do you have any explanation for that?

  **Our Response:**
  We made a terrible mistake in the original manuscript that the first guess for the next cycling step was not updated correctly. This mistake was fixed in the original manuscript, and the average first guess forecast errors are decreased with increasing number of DA cycles. To be specific, in most cases, FFS tends to increase with increasing number of DA cycles, and MAE tends to decrease with increasing number of DA cycles. (Figure 9, 10, 13, 14)

- Why do you use quality control to control the number of assimilated observations? In my view, quality control should sort out erroneous or non-representative observations. You should rather control the number of observations by horizontal localization, thinning and superobbing as well as observation error

**Our Response:**

We think there is a misunderstanding here. We did not use the quality control method to control the number of assimilated observations. In fact, it is the DART built-in approaches that reject some of the observations because assimilating these observations could bring detrimental impacts. The observations were rejected either for non-monotonic pressure (which is not physically consistent and the RTTOV returns a default value of -88888) or for the points where differences between the observations and the prior ensemble mean equivalent observations are too large. If too large adjustments were made to the model state variables, large discontinuity may cause the collapse of WRF model. As a result, some of the observations were not effectively assimilated during the assimilation process. Therefore, the focus of this part is not the quality control for the observations, but the observation utilization during the assimilation which has nothing to do with the "quality" of the observations.

It is true that the horizontal localization, thinning and superobbing, and observation error also controls the number of observations. Therefore, relevant experiments (superobbing was not discussed in this study) were added in the revised manuscript. (Table 1, L304; Section 3.2, L537-548)

- If I understand correctly, the outlier threshold acts on the first guess departure of the ensemble mean. Large first guess departures typically occur when clouds are missing or you have location error of clouds which tends to happen quite often for clouds- and precipitation-sensitive observations. In my opinion, being able to correct for such location errors or false alarms in the analysis is of particular importance. Why do you choose to sort out these observations? Would it be possible to inflate observation error in those cases rather than not assimilating the observations at all?

**Our Response:**

It is true that large first guess departures contain critical cloud information, and therefore, assimilating these observations has great potentials to improve the analysis and first-guess forecast. A method to use these observations is to increase the outlier threshold value. The beneficial impacts of utilizing large first guess departures were validated in section 3.2.1 (L516-536).

We think assimilating observations corresponding to the large first guess departures is beneficial to remove the false alarm clouds, but cannot improve the underestimated clouds due to zero-spread problem for current WRF/DART system.

We did not try to sort these observations. On the contrary, we try to utilize as many observations as possible by setting a larger outlier threshold value. Positive impacts were revealed for the analysis and first-guess forecast for the cycled DA experiments with larger outlier threshold value.

**Sec 5.**

- Updating only cloud variables for NWP forecasts is not practical for operational NWP. Do you have other suggestions to deal with potential detrimental effects on forecast skill of temperature, humidity, wind etc.?

  **Our Response:**
  We correct some terrible mistakes in the original manuscript. In the revised manuscript, positive impacts were revealed for the analysis and first-guess forecast of non-cloud variables. Including the non-cloid variables improved the analysis and first-guess forecast quality for the cycled DA experiments. Therefore, both the cloud and non-cloud variables should be updated for better forecasts.(L470-515)

- Please replace "modelling experiments" by "data assimilation experiments". As far as I understood you do not show forecasts or their verification.

  **Our Response:**
  Corrected. We used the "cycled data assimilation (DA) experiments" throughout the manuscript.

- Please replace "cloud simulations" by "cloud states". You mostly look at analysis ensemble means which is not the same as a "simulation" or a "free model state"

  **Our Response:**
  Corrected. We used the "cloud state variables" throughout the manuscript.

- If you verify analyses the term "cloud forecasting skills" may be inappropriate

  **Our Response:**
  Corrected. In the revised manuscript, both the analysis the first-guess forecast were analyzed.

- Why would increased model grid spacing lead to a more nonlinear relationship between radiance and LWP / IWP? Or do you mean nonlinearity due to resolved convective processes? Please clarify

  **Our Response:**
  Here the nonlinearity denotes the nonlinear relationship between cloud variables (such as CWP) and VIS radiance. Existing studies indicate that 3D radiative effects could be strengthened due to the decrease of grid spacing. The 3D radiative effects

could make the nonlinear relationship between CWP and VIS radiance more complicated. (L621-623)

- Please elaborate a bit more on ideas for further research based on your found results in the conclusion. What did you learn and what do you suggest to deal with the found problems? What do you suggest for operational VIS radiance data assimilation?

**Our Response:**
Further studies should be extended to cloud-resolving model simulations like Scheck et al. (2020) to fully take the advantage of high-resolution satellite VIS radiance data. Other future works should include 1) Optimization of forward operators and estimation of observation errors, 2) Improvements on computational cost and accuracy of forward operators, 3) Correction of errors due to non-Gaussian and non-linear problems, 4) Techniques to reduce the cloud location errors. (L620-652)

**Spelling, grammar, typos**

- L-8: *there are great potentials in assimilating* change to: *there is great potential related to assimilating …*

**Our Response:**
Corrected. (L48)

- L-31: unique cloud information complementing the one contained in IR and MW data

**Our Response:**
Corrected. (L50-51)

- L-40: direct data assimilation critically depends on observation operators

**Our Response:**
Corrected. (L65)

- L-47: Method for Fast Satellite Image Synthesis (MFASIS)

**Our Response:**
Corrected. (L73)

- L-53: single-scattering method for SW radiative processes

**Our Response:**
Corrected. (L78)

- L-59, 61, 64: kill superflues "the": assimilated GOES-9 VIS radiance, is ensemble-based methods

**Our Response:**
Corrected. (L84, 89)

- L-93: revise end of sentence "A nature run is.."

**Our Response:**
Corrected. (L129-131)

- L-140: Other parameters not explicitly mentioned are set to default values.

**Our Response:**
Corrected. (L211)

- L-186: "To demonstrate the basic ability of the DA scheme.:"; it is unclear what you mean by that. Do you mean "to demonstrate the basic technical functionality of assimilating visible radiance data by employing EAKF"?

**Our Response:**
Yes it is. Thank you for this suggestion. Corrected. (L266-267)

- 281: double "the"

**Our Response:**
Corrected. (L411)

- L-302: could clearly suppress false alarm clouds

**Our Response:**
Corrected. (L435-436)

- L-324: through spurious correlation between VIS radiance

**Our Response:**
Corrected throughout the manuscript (L336, 346, 374).

- L-339: At the initial cycling step, convective initiation occurred in the nature run

**Our Response:**
Since the experiment set-ups were changed in the revised manuscript, this part was deleted.

- L-348: non-cloud state variables obtain analysis increments with wrong sign such that analysis error is increased compared to first guess error

**Our Response:**
Since the experiment set-ups were changed in the revised manuscript, this part was changed as well.

- L-363: false alarm clouds

**Our Response:**
Corrected throughout the manuscript. (L325, L360, L376, L435-436)

- L-364: much closer

**Our Response:**
Since the experiment set-ups were changed in the revised manuscript, this part was changed as well.

- L-396: by the DART system

**Our Response:**
Corrected throughout the manuscript.

- L-408: by the detrimental effects on analysis error of the non-cloud …

**Our Response:**
Since the experiment set-ups were changed in the revised manuscript, this part was changed as well.

- L-420: such as the Atmospheric Motion Vector

**Our Response:**
Since the experiment set-ups were changed in the revised manuscript, this part was deleted.

- L-426: life cycle, i.e. the intensification and decay processes of a cyclone

**Our Response:**
Since the experiment set-ups were changed in the revised manuscript, this part was changed as well.

- L-428: the adjustment of CWP

**Our Response:**
Since the experiment set-ups were changed in the revised manuscript, this part was changed as well.

- In general: replace thermodynamic by non-cloud

**Our Response:**
Corrected throughout the manuscript.

---

## Author Response (AR2)

**Response to Reviewer #2**

**General Comment:**
I recommend you to fundamentally rework the text - potentially with the help of an English language expert.

**Our response:**
The manuscript was polished very carefully to avoid typo errors and misleading due to improper description. We hope that the revised manuscript could meet the standard of GMD for publishing.

**Remark:**
Please find a few closing remarks from my side:
* Replace „assimilation" by „data assimilation"

**Our response:**
Corrected. Sometimes "data assimilation" is abbreviated as "DA" for simplicity. The revisions are marked in red in the revised manuscript (L32 and L292 for "data assimilation". Throughout the manuscript for "DA")

**Remark:**
* Replace "impacts" by "impact". There is no plural
**Our response:**
Corrected throughout the manuscript. The revisions are marked in red in the revised manuscript.

**Remark:**
* Revise all articles („the") carefully.
**Our response:**
Corrected.

**Remark:**
* Re-check all your prepositions: e.g. "compared to" instead of "compared with"
**Our response:**
I searched the difference between "compared to" and "compared with". Here is a representative answer
(https://www.dailywritingtips.com/compared-to-or-compared-with/):
*"compare to" is to point out or imply resemblances between objects regarded as essentially of a different order;*
*"compare with" is mainly to point out differences between objects regarded as essentially of the same order.*
*Thus, life has been **compared to** a pilgrimage, to a drama, to a battle; Congress may be **compared with** the British Parliament.*

In our manuscript, both comparisons stress the differences between two items. Therefore, we think "Compared with" is more appropriate than "Compared to". (L69, 373, 425, 542)

**Remark:**
* Re-check all your figure captions for completeness and understandability
**Our response:**
All checked. We added subcaptions for Figures 2, 3, 9, 10, 13, 14, 16. In addition, Figures 1, 6, and 12 were revised. The rvisions were marked in red in the revised manuscript.

**Remark:**
* Please correct: FSS measures the spatial accuracy of a spatially inhomogeneous variable on a certain spatial scale. Thereby it mitigates the double penalty problem
**Our response:**
Corrected. (L309-311)

**Remark:**
* Add the verification of short-range forecasts in the outlook. I highly appreciate that you have included first guess forecast verification. However, the DA is actually targeted at improving cloud forecasts and related quantities during longer-term forecasts
**Our response:**
Thank you for pointing this out. On one hand, the limited impact of the DA on rain rate is partly caused by the shortcomings of the DA procedure, including the inabilities of the DA to improve cloud vertical structures and cloud phases. On the other hand, the rain rate is influenced by the spin-up effects. The spin-up effects may introduce false alarm precipitation due to the interactions between the model dynamics and microphysics when smaller scales are now well represented in the initial conditions and lateral boundary conditions (Short and Petch, 2022). In this study, we started to assimilate the synthetic FY-4A visible radiance data after 2-hour cold start and run the model for 10-hour forecast (02:00 ~ 12:00 UTC). This is a too short time period to exclude the spin up. Improvements on precipitation should be expected for longer forecasts. (L641-647)

**Remark:**
* A comment on why precipitation may not be improved in your forecasts.
1. Precipitation is subject to strong spin-up effects: you may observe improvements in longer forecasts
**Our response:**
Corrected. (L641-647)

**Remark:**
2. In general I suppose that improvement of horizontal location error of cloud fields

can subsequently lead to better spatial precipitation accuracy - at least in many cases, e.g. deep convection where vertical cloud structure is less important since the cloud may extend from boundary layer up to troposphere

**Our response:**
We agree with you that improvement on horizontal cloud location errors could improve precipitation forecasts in some cases. For example, the vertical cloud structure is less important for some deep convection since the cloud may extend from boundary layer up to troposphere (Hu et al., 2021). (L638-640)

**Remark:**
3. In my experience, due to a) the strong dependence of reflectance on subgrid-scale parameterized clouds and b) the strong interaction of VIS radiances with model biases and c) the fact that nearly every process related to clouds, precipitation and radiation is subject to parameterization and potentially compensating model error often requires a well-tuned model or additional model tuning to really obtain a positive impact on precipitation forecasting from better cloud fields

**Our response:**
It is true that the NWP model errors on cloud and precipitation forecasts should be considered in the DA processes for the real FY-4A visible radiance DA. On one hand, the parameterization (such as the subgrid-scale cloud fraction parameterization in this study) is closely related to the calculation of synthetic visible radiance by a forward operator. On the other hand, the formation and dissipation of cloud and precipitation highly depend on the model parameterization. Suboptimal parameterization may introduce large model bias in some cases due to unsolved scales and processes (Janjić et al., 2021). The model bias could introduce negative impact on cloud and precipitation forecasts. Therefore, the NWP model should be tuned to properly represent the scale-dependent microphysical processes in order to fully realize the effects of the FY-4A visible radiance DA. (L648-655)

**Other changed not explicitly mentioned are marked in red.**